# Quantum Autoencoder: An Efficient Representation Learner for Quantum Features

## Abstract

Quantum machine learning methods often rely on fixed, hand-crafted quantum encodings that may not capture optimal features for downstream tasks. In this work, we study the power of quantum autoencoders in learning data-driven quantum representations. We first theoretically demonstrate that the quantum autoencoder method is efficient in terms of sample complexity throughout the entire training process. Then we numerically train the quantum autoencoder on 3 million peptide sequences, and evaluate their effectiveness across multiple peptide classification problems, including antihypertensive activity, blood–brain barrier-penetration, and cytotoxicity. The learned representations were compared against Hamiltonian-evolved baselines using a quantum kernel with support vector machines. Results show that quantum autoencoder learned representations achieve accuracy improvements ranging from 0.4% to 8.1% over Hamiltonian baselines across seven datasets, demonstrating effective generalization across biologically distinct datasets, with pre-training enabling effective transfer learning without task-specific fine-tuning. This work establishes that quantum autoencoder architectures can effectively learn from large-scale datasets (3 million samples) with compact parameterizations ($\sim$900 parameters), demonstrating their viability for practical quantum applications.

## 1 Introduction

Quantum machine learning (QML) has emerged as a promising field that integrates principles of quantum computing with classical machine learning objectives, offering the potential for enhanced expressivity and improved pattern recognition in high-dimensional learning tasks (Biamonte et al., 2017). Central to QML model performance is the encoding of classical data into quantum states, a critical preprocessing step that fundamentally determines the model's representational capacity and generalization ability (Huang et al., 2021a; Schuld et al., 2021). Whether in quantum support vector machines (QSVMs) (Rebentrost et al., 2014; Wu et al., 2023b; Havlíček et al., 2019), variational quantum classifiers (VQCs) (Cerezo et al., 2021), or quantum generative models (Dallaire-Demers & Killoran, 2018; Carrasquilla et al., 2019; Gao et al., 2018), the choice of quantum state representation is often a key bottleneck for the downstream performance.

In practice, most QML pipelines rely on fixed, task-agnostic encodings to convert classical data into quantum states. Common strategies include *angle encoding* (Schuld & Killoran, 2019), where scalar features are mapped to qubit rotations; *amplitude encoding* (Mottonen et al., 2004), which encodes normalized feature vectors into the amplitude of quantum states; and *Hamiltonian evolution* (Wecker et al., 2015; Wiersema et al., 2020), where classical data is used to parametrize a system Hamiltonian and the resulting time-evolved quantum state serves as the data representations. These approaches are appealing due to their computational simplicity and circuit efficiency. However, they suffer from a key limitation: they are static and unable to adapt to the statistical structure of specific datasets or optimize for the downstream learning task. In contrast, classical machine learning has achieved major performance gains through the learned representations enabled by deep learning architectures, including unsupervised pre-training models like protein language models (PLMs) that are specifically adapted to both the data distribution and downstream tasks (Gelman et al., 2025). The quantum field has yet to fully embrace this data-driven paradigm. Most current quantum encodings remain

manually designed and disconnected from the structure of the input data or the requirements of the learning task, potentially limiting the effectiveness of QML models in practice.

This contrast has motivated increasing interest in quantum analogs of representation learning. Among these, the quantum autoencoder has been proposed as a variational circuit architecture that learns to compress quantum states by discarding redundant qubits. While quantum autoencoders were originally developed for tasks such as quantum state compression, they offer an appealing framework for learning quantum representations for classical data. Despite this, their role in supervised QML pipelines, especially in learning general-purpose representations for downstream classification, remains underexplored. Moreover, the scalability of quantum autoencoder training on large classical datasets encoded into quantum states has not been explored.

A key motivation for this work is that quantum data encoding is one of the key factors govering quantum machine learning models can achieve any practical advantage. Recent theoretical results show that well-structured encodings can provably yield quantum–classical separations (Liu et al., 2021; Huang et al., 2021a), whereas poorly chosen or random encodings are often classically simulable or lead to exponentially suppressed quantum kernels (Bermejo et al., 2024; Angrisani et al., 2024; Thanasilp et al., 2024). Consequently, the design of effective encodings has emerged as a central challenge for practical QML, and current hand-crafted or fixed encodings may not fully capture the statistical structure of complex datasets such as biological sequences.

A key motivation for this work is that quantum data encoding is one of the key factors that determine whether QML models can achieve any practical advantage. Recent theoretical results show that well-structured encodings can provably yield quantum–classical separations (Liu et al., 2021; Huang et al., 2021a), whereas poorly chosen or random encodings are often classically simulable or result in exponentially small kernel values (Bermejo et al., 2024; Angrisani et al., 2024; Thanasilp et al., 2024). Consequently, the design of effective encodings has emerged as a central challenge for practical QML, and current hand-crafted or fixed encodings may not fully capture the statistical structure of complex datasets such as biological sequences, whose underlying structure is not well understood.

We train quantum autoencoders with different compression schemes and circuit depths, then extract the learned compressed quantum states as fixed representations for each input from the downstream datasets. These representations are evaluated using support vector machines (SVMs) (Cortes & Vapnik, 1995) with trace distance kernels. As a baseline, we compare against Hamiltonian evolved quantum states encoded from the same downstream datasets. This setup allows us to isolate the impact of the learned representation on classification performance while keeping the downstream classifier and data consistent.

Our results show that quantum autoencoder learned representations obtained measurable improvements of 0.4% to 8.1% in test accuracy over Hamiltonian-based encodings across all tasks, while also demonstrating strong transferability: models trained in an unsupervised fashion on large-scale data generalize effectively to biologically distinct downstream datasets without requiring fine-tuning. This indicates that quantum autoencoders act as useful pretrained models, giving quantum computing a counterpart to successful classical representation learning methods.

Within this context, our work provides both large-scale empirical evidence (pretraining on 3M sequences) and theoretical justification that a quantum autoencoder can learn an improved encoding relative to a fixed Hamiltonian map. Even though the empirical gains are moderate under the highly compressed 8–10 qubit setting considered here, the results demonstrate that quantum feature maps can be enhanced through data-driven learning rather than manual specification. This positions the quantum autoencoder as a practical approach for learning adaptive quantum encodings and motivates further exploration of scalable quantum representation learning.

## 2 RELATED WORK

### 2.1 QUANTUM DATA ENCODING

Encoding classical data into quantum states is a fundamental step in QML pipelines, as it determines the subspace of the Hilbert space explored by the models. Common approaches include:

- **Angle encoding:** For a data vector $\boldsymbol{x}_i = (x_{i,1}, x_{i,2}, \ldots, x_{i,n})$, each scalar value $x_{i,j}$ is used to parametrize a single-qubit rotation (Schuld & Killoran, 2019),

$$|0\rangle^{\otimes n} \mapsto \bigotimes_{j=1}^{n} R_y(x_{i,j}) |0\rangle = \bigotimes_{j=1}^{n} \left( \cos(x_{ij}/2)|0\rangle + \sin(x_{ij}/2)|1\rangle \right).$$

- **Amplitude encoding:** The normalized vector $\boldsymbol{x}_i \in \mathbb{R}^{2^n}$ is encoded directly into the amplitudes of a quantum state (Mottonen et al., 2004),

$$\boldsymbol{x}_i \mapsto |\psi(\boldsymbol{x}_i)\rangle = \sum_{j=0}^{2^n-1} x_{i,j}|j\rangle, \quad \|\boldsymbol{x}_i\|_2 = 1.$$

- **Hamiltonian evolution encoding:** The vector $\boldsymbol{x}_i$ parametrizes a Hamiltonian $H(\boldsymbol{x}_i)$, and the encoded state is obtained by time evolution (Wecker et al., 2015; Wiersema et al., 2020),

$$|\psi(\boldsymbol{x}_i)\rangle = e^{-iH(\boldsymbol{x}_i)t} |0\rangle^{\otimes n},$$

where $t$ denotes the evolution time.

These encodings have been central to early QML experiments because of their simplicity and relatively low circuit depth. In this work, we selected the Hamiltonian evolution encoding as a fixed baseline against which to compare the learned, data-driven quantum representation.

## 2.2 QUANTUM AUTOENCODERS

Quantum autoencoders are variational quantum circuits designed to efficiently compress quantum states by removing redundant degrees of freedom while retaining the information necessary to reconstruct the original state (Romero et al., 2017). The typical architecture partitions the qubits into a latent register, which retains compressed information, and a trash register, which is discarded after compression. The parameterized unitary is trained to maximize the fidelity between the trash register and a reference state via a SWAP test (Buhrman et al., 2001).

Originally, quantum autoencoders were proposed for efficient compression of quantum data (Romero et al., 2017). Subsequent work has extended quantum autoencoders to denoising quantum data (Bondarenko & Feldmann, 2020) and experimental demonstrations with photonic systems (Pepper et al., 2019), where data compression has been realized using photons.

In molecular domains, a quantum variational autoencoder with spherical latent variable learning has been applied to 3D molecule generation, where the goal is to learn latent variables that capture molecular geometry and enable the synthesis of novel molecular structures (Wu et al., 2024). In parallel, a quantum autoencoder framework was proposed for molecular representation learning, focusing on compressing SMILES-based molecular encodings into a latent quantum register (Pan et al., 2025). Their evaluation was based on the reconstruction fidelity, demonstrating that compressed states could retain information about the original input. However, the learned representations were not applied to downstream supervised tasks such as property prediction, leaving their practical usefulness for machine learning unexplored.

Beyond compression-oriented designs, the quantum variational autoencoder (Khoshaman et al., 2018) extends the autoencoder concept to generative modeling. In this framework, the latent distribution is parameterized by a quantum Boltzmann machine, enabling sampling from quantum-enhanced latent variables. Unlike the quantum autoencoder, which is primarily focused on compressing quantum states into a reduced register, the quantum variational autoencoder shifts the focus toward modeling quantum-enhanced generative processes.

In contrast, our work explicitly evaluates quantum autoencoder representations in peptide classification tasks, benchmarking against Hamiltonian baselines. This direct use of learned representations for prediction differentiates our study, showing not only that the quantum autoencoder can compress quantum states but also that the resulting representations improve downstream performance.

## 2.3 REPRESENTATION LEARNING FOR SEQUENCE DATA

The success of machine learning algorithms generally depends on the choice of data representation, which can hide or reveal the different explanatory factors of variation behind the data (Bengio et al.,

2013). The learned representations have driven major advances in machine learning by automatically learning features that capture richer patterns than manual engineering and enable effective transfer learning.

The bioinformatics field has begun to adopt this approach in recent years. Early computational approaches to peptide and protein modeling relied on fixed encodings such as amino acid composition or physicochemical descriptors. While useful in certain contexts, these handcrafted features were limited in their ability to capture the complex dependencies that govern biological sequence-function relationships. More recently, PLMs such as ESM, ProtBERT, UniRep (Lin et al., 2023; Elnaggar et al., 2021; Alley et al., 2019), which are trained in an unsupervised (more precisely, self-supervised) manner on millions of sequences, as well as PeptideBERT, which fine-tunes ProtBERT for peptide property prediction tasks including hemolysis and solubility (Guntuboina et al., 2023). These models demonstrate that unsupervised pre-training on large unlabeled datasets can produce general-purpose representations that transfer effectively to supervised prediction problems.

Most existing QML literature has focused on demonstrating theoretical quantum advantage or on developing efficient state preparation methods that exploit symmetries or well-defined mathematical structures (Huang et al., 2022; 2021b), whereas in many real-world problems the underlying structure is unclear or poorly defined. For example, in biological sequence modeling, the relationship between peptide or protein sequence and function is highly complex and remains poorly understood, making it difficult to capture with fixed encodings. This motivates the need for adaptive, data-driven quantum representation learning approaches, such as quantum autoencoders, which can make use of large unlabeled datasets to learn representations that could potentially improve downstream task performance.

## 3 METHODS

Our workflow consists of two main stages:

(1) unsupervised pre-training of a quantum autoencoder on Hamiltonian encoded quantum states from 3 million unlabeled peptide sequences;

(2) downstream evaluation of the learned representation on seven peptide supervised classification tasks.

The Hamiltonian encoded states play two roles: (i) they serve as the input to the quantum autoencoder; (ii) they are also used to construct a kernel that trains classical SVMs to provide a baseline for comparison. The quantum autoencoder compresses quantum states by discarding trash qubits, with the remaining subsystem serving as the learned representation. These representations are then used to construct a kernel for SVMs classification. A schematic of the overall workflow is shown in Fig. 1.

### 3.1 REPRESENTATION PIPELINES

All experiments used the same peptide datasets, but the initial data representation varied by method. Quantum approaches (the quantum autoencoder and Hamiltonian baseline) and classical baselines started from one-hot encoding matrices, where rows correspond to amino acid type and columns to sequence positions. These matrices were flattened into binary vectors for subsequent processing. On the other hand, PLMs operated directly on raw amino acid sequences, applying their own tokenization before producing representations. Detailed dataset collection, preparation, and preprocessing steps are provided in Appendix E.1 (pre-training) and Appendix E.2 (downstream).

**Hamiltonian encoding baseline.** The flattened one-hot vectors parametrized a system Hamiltonian, and the resulting time-evolved states were used directly to construct a kernel for support vector machines (SVMs). This serves as the fixed quantum encoding baseline (see Section 3.4.2 for kernel construction details).

**Quantum autoencoder.** The same Hamiltonian-encoded states were provided as input to the quantum autoencoder, which compressed them by tracing out designated trash qubits. The remaining subsystem was treated as the learned representation. Reduced density matrices from this subsystem were then used to construct kernels for SVM classification (see Section 3.4.1 for details).

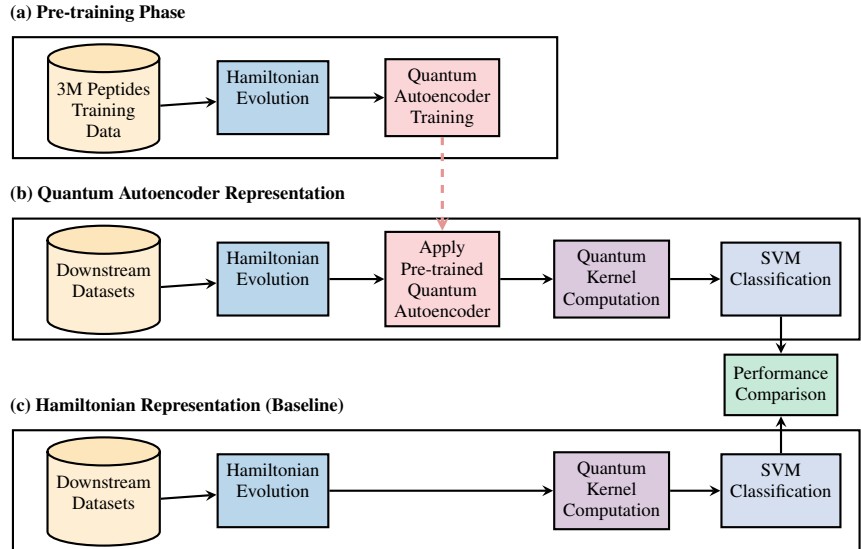

Figure 1: Methodology Overview: Quantum Autoencoder vs Hamiltonian Baseline Comparison. (a) Pre-training on 3M peptide sequences to learn quantum representations, (b) using pre-trained quantum autoencoders for downstream tasks, and (c) using direct Hamiltonian evolution as baselines. Both (b) and (c) employ quantum trace distance as a kernel for SVM classification, enabling direct performance comparison.

**Classical baselines.** The flattened one-hot vectors were used as input to five candidate classifiers: SVM with an RBF kernel, SVM with a linear kernel, random forest (Breiman, 2001), XG-Boost (Chen & Guestrin, 2016), and logistic regression (McCullagh, 2019). For each dataset, the best-performing model was selected based on cross-validated accuracy.

**Protein language models (PLMs).** Raw amino acid sequences were provided as input to pre-trained PLMs, including ESM, ProtBERT, and related models (Lin et al., 2023; Elnaggar et al., 2021). These models applied their own tokenization and generated sequence representations, which were subsequently evaluated with the same set of five classical classifiers used for the classical baselines. The best performing PLM-classifier combination was reported for each dataset, and the full list of PLMs considered is provided in Appendix D.

This setup enables two types of comparisons. In the quantum setting, we evaluate whether quantum autoencoder representations provide improvements over fixed Hamiltonian encoding baselines. In the classical setting, we evaluate whether pre-trained PLMs provide improvements over traditional one-hot encodings when combined with classical classifiers. These paired comparisons focus on the relative contribution of learned versus fixed representations in each domain, with all classifiers evaluated under 5-fold cross-validation to ensure fairness and consistency, where hyperparameter tuning was conducted within the cross-validation loop for models with tunable parameters to prevent data leakage and ensure robust model selection.

## 3.2 QUANTUM AUTOENCODER

The quantum autoencoder consists of a parametrized encoder circuit that compresses the input state by tracing out some trash qubits, leaving the remaining subsystem as the learned representation. During training, for each one-hot vector $\boldsymbol{x}_i \in \mathcal{D}$, we first map the classical data $\boldsymbol{x}_i$ to $n$-qubit Hilbert space $|\psi(\boldsymbol{x}_i)\rangle$ by using the Hamiltonian encoding method. Then, a $n$-qubit, $d$-layer variational quantum circuit $U(\vec{\boldsymbol{\theta}})$ is performed on $|\psi(\boldsymbol{x}_i)\rangle$, followed by a computational basis measurement to the $m$ trash qubits, where $m = \mathcal{O}(1)$ and $m \ll n$. The probability of obtaining all $|0\rangle$ outcomes is used as the loss function, that is

$$\mathcal{L}(\vec{\boldsymbol{\theta}}) = \mathrm{Tr}\left[\Pi_m U(\vec{\boldsymbol{\theta}})\mathbb{E}_{\boldsymbol{x}_i \sim \mathcal{D}}\left(|\psi(\boldsymbol{x}_i)\rangle\langle\psi(\boldsymbol{x}_i)|\right)U^\dagger(\vec{\boldsymbol{\theta}})\right], \tag{1}$$

where the $m$-qubit local observable $\Pi_m = |0\rangle\langle 0|^{\otimes m}$. The encoder parameters $\vec{\theta}$ are optimized to maximize the probability $\mathcal{L}(\vec{\theta})$, effectively pushing information away from the trash subsystem and into the remaining subsystem, which then serves as the learned representation. Notably, we only train the encoder in the training process, which is mathematically equivariant to train both encoder and decoder when they share the same quantum circuit. Specifically, let $\theta^* = \arg\max_{\vec{\theta}} \mathcal{L}(\vec{\theta})$, the learned quantum feature state (density matrix) is

$$\phi(\boldsymbol{x}_i) = \mathrm{Tr}_m \left[ U(\boldsymbol{\theta}^*)|\psi(\boldsymbol{x}_i)\rangle\langle\psi(\boldsymbol{x}_i)|U^\dagger(\boldsymbol{\theta}^*) \right]. \tag{2}$$

The two theorems below provide theoretical analysis on the efficiency aspect of the quantum autoencoder method. The proofs of these theorems are given in Appendix A and Appendix B, respectively.

**Theorem 1.** *Given the Hamiltonian encoding method $|\psi(\boldsymbol{x}_i)\rangle = e^{-iH(\boldsymbol{x}_i)t}|0^n\rangle$, where the Hamiltonian $H(\boldsymbol{x}_i) = \sum_{s=1}^{L} \boldsymbol{x}_i(s)P_s$, and the evolution time $t = \mathcal{O}(1)$. Here, $P_s$ represents a 'constant-weight' Pauli operator non-trivially acting on constant number of qubits. Then a quantum circuit $U(\vec{\theta})$, whose depth $d$ satisfies*

$$d \leq \mathrm{poly}\log(nt/\epsilon),$$

*suffices to implement the quantum autoencoder task.*

**Theorem 2.** *For any quantum autoencoder task with loss function $\mathcal{L}(\vec{\theta})$,*

$$N = \mathcal{O}\left( \frac{3^{\log(1/\epsilon\delta)} \log(1/\epsilon\delta) \log(n)}{\epsilon^2} \right) \tag{3}$$

*samples drawn from $\mathbb{E}_{\boldsymbol{x}_i \sim \mathcal{D}} [|\psi(\boldsymbol{x}_i)\rangle\langle\psi(\boldsymbol{x}_i)|]$ suffice to estimate $\mathcal{L}(\vec{\theta})$ throughout the training process, with success probability $\geq 1 - \delta$. In other words, $N$ samples are sufficient to guarantee an $\epsilon$-approximation to $\mathcal{L}(\vec{\theta})$ for all candidate variational parameters $\vec{\theta}$.*

Our first result focuses on the quantum circuit depth of $U(\vec{\theta})$ used in the quantum autoencoder. In general, training the variational quantum circuit $U(\vec{\theta})$ would be classically hard in the worst-case scenario (Bittel & Kliesch, 2021). Here, we rigorously prove that the sample complexity in training the quantum autoencoder is efficient, which can be summarized as theorem 2. The overall structure of the quantum autoencoder circuit is illustrated in Fig. 2.

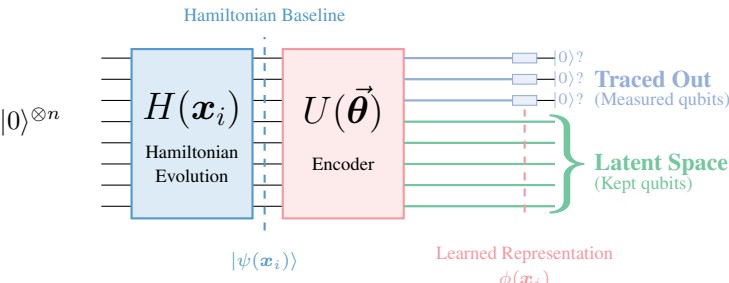

Figure 2: Quantum autoencoder architecture for unsupervised representation learning. The circuit processes input peptide sequences in two stages: (1) Hamiltonian evolution $H(\boldsymbol{x}_i)$ encodes classical data into quantum states, and (2) a parameterized encoder $U(\vec{\theta})$ performs dimensional compression. The encoder discards qubits (blue) while preserving latent information (green), producing compressed quantum representations $\phi(\boldsymbol{x}_i)$ for downstream peptide classification tasks.

### 3.3 PRE-TRAINING SETUP

Training was performed on a large-scale dataset of 3 million peptide sequences, represented as flattened one-hot vectors and then mapped into quantum states via Hamiltonian evolution. We used mini-batch training with a batch size of 1024, optimizing with the Adam optimizer (Kingma & Ba, 2015) at a learning rate of $10^{-2}$. Models with 8 input qubits were trained for up to 10 epochs, while

models with 10 input qubits were trained for up to 5 epochs. In practice, all models converged within 1-2 epochs, so the larger maximum epoch limits simply provided a safety margin and did not affect the final results. To ensure fairness, we trained 18 quantum autoencoder variants covering all combinations of input size (8 or 10 qubits), circuit depth (10, 20, or 30 layers), and number of traced-out qubits (1, 2, or 3). The best-performing variant for each dataset was selected based on cross-validation accuracy, analogous to hyperparameter optimization in classical machine learning. The Hamiltonian baseline, by contrast, involves no trainable parameters and serves as a fixed reference encoding, but was evaluated under the same cross-validation splits for comparability. Our aim in training multiple models was not to analyze architectural effects individually, but to ensure coverage of a representative model space. This setup provides a fair test of whether the quantum autoencoder can successfully learn from large-scale peptide data and produce representations that outperform fixed Hamiltonian encodings.

### 3.4 DOWNSTREAM EVALUATION

#### 3.4.1 QUANTUM AUTOENCODER KERNEL

Once trained, the quantum autoencoder provides compressed representations obtained from the unmeasured subsystem of the quantum state. To evaluate these representations in supervised learning tasks, we construct a similarity kernel as follows. First, pairwise trace distances are computed between the reduced density matrices corresponding to different samples. These distances are then converted into similarities by applying $k_{\mathrm{QAE}}(\boldsymbol{x}_i, \boldsymbol{x}_j) = 1 - \mathrm{dist}_{\mathrm{tr}}^2(\boldsymbol{x}_i, \boldsymbol{x}_j)$, where the trace distance

$$\mathrm{dist}_{\mathrm{tr}}(\boldsymbol{x}_i, \boldsymbol{x}_j) = \frac{1}{2}\|\phi(\boldsymbol{x}_i) - \phi(\boldsymbol{x}_j)\|_1 \tag{4}$$

and the quantum feature state $\phi(\boldsymbol{x}_i)$ (a $(2^{n-m} \times 2^{n-m})$-sized density matrix) is given by equation 2. Generally, estimating the trace distance between mixed states are even quantum hard. However, the condition $m \ll n$ enables $\phi(\boldsymbol{x}_i)$ being a low-rank density matrix for $i \in [N]$, where $N$ represents the number of data $\boldsymbol{x}_i$ within the data set $\mathcal{D}$, and the related trace distance can be efficiently estimated with the sample complexity of $\mathcal{O}(\mathrm{rank}^2(\phi(\boldsymbol{x}_j))/\epsilon^5)$, where $\epsilon$ represents the additive error (Wang & Zhang, 2023).

#### 3.4.2 HAMILTONIAN BASELINE KERNEL

This quantum autoencoder based kernel is compared against a baseline kernel constructed directly from Hamiltonian-encoded states without training. Specifically, suppose $\mathcal{D} = \{(\boldsymbol{x}_i)\}_{i=1}^N$ representing the flattened one-hot vectors, the Hamiltonian encoding method achieves the map $\boldsymbol{x}_i \mapsto |\psi(\boldsymbol{x}_i)\rangle = e^{-iH(\boldsymbol{x}_i)t}|0^n\rangle$ (Zhuang et al., 2024). Here, we suppose the one-hot vector $\boldsymbol{x}_i \in \mathbb{R}^L$ represents a $L$-dimensional real vector, the involved local Hamiltonian $H(\boldsymbol{x}_i) = \sum_{s=1}^L \boldsymbol{x}_i(s)P_s$, and $P_s$ represents a 'constant-weight' Pauli operator non-trivially acting on a constant number of qubits. The quantum kernel function is defined as $k_H(\boldsymbol{x}_i, \boldsymbol{x}_j) = |\langle\psi(\boldsymbol{x}_i)|\psi(\boldsymbol{x}_j)\rangle|^2$, and it can be computed efficiently by using the SWAP-test method (Buhrman et al., 2001).

In both cases, the resulting kernel matrices are used to train classical SVMs for downstream classification. Importantly, the same trace distance kernel is used for both the Hamiltonian baseline and the learned representations; the only difference lies in the quantum states used to compute the kernel entries. When the encoding produces pure states, i.e. $\psi(\boldsymbol{x}) = |\psi(\boldsymbol{x})\rangle\langle\psi(\boldsymbol{x})|$, the trace-distance kernel $k_{\mathrm{QAE}}(\boldsymbol{x}_i, \boldsymbol{x}_j) = 1 - \mathrm{dist}_{\mathrm{tr}}^2(\boldsymbol{x}_i, \boldsymbol{x}_j)$ reduces to $k(\boldsymbol{x}_i, \boldsymbol{x}_j) = |\langle\psi(\boldsymbol{x}_i)|\psi(\boldsymbol{x}_j)\rangle|^2$. The classification accuracy obtained with the Hamiltonian baseline serves as the reference point for measuring improvements achieved by the learned representations. This procedure benchmarks whether unsupervised quantum representation learning can provide advantages over fixed encodings in peptide sequence classification. We also provide in Appendix C a generalization-theoretic justification for why the quantum autoencoder, as a completely positive trace-preserving (CPTP) map, can reduce trace distances and improve kernel-based generalization, together with the full proof.

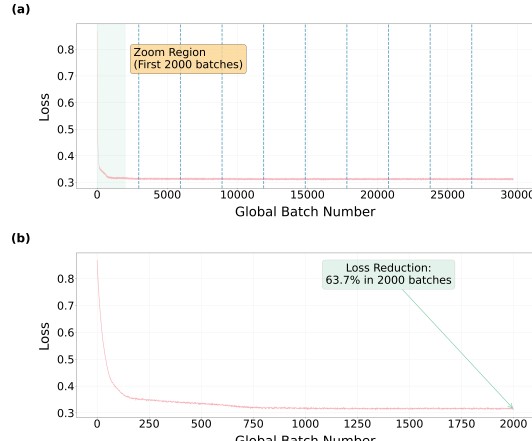

Figure 3: Training loss reduction during quantum autoencoder pre-training. (a) Complete training trajectory showing loss versus global batch number across all epochs, with the zoom region highlighted in green covering the first 2000 batches where rapid convergence occurs. Blue vertical dashed lines indicate epoch boundaries. (b) Detailed view of the first 2000 batches showing the rapid loss reduction, where the model achieves 63.7% loss reduction. The loss curve demonstrates fast initial convergence followed by stabilization.

## 4 NUMERICAL SIMULATION RESULTS

### 4.1 QUANTUM AUTOENCODERS TRAINING

Training of the quantum autoencoders converged rapidly across all 18 variants. Figure 3 shows the loss trajectory for one representative model. More than 60% of the total loss reduction occurs within the first 2000 batches, corresponding to less than a single epoch over the 3 million peptide corpus. After this initial phase, the training loss quickly stabilizes at a low value and remains consistent throughout the remainder of training.

Across all architectures, final training losses ranged from approximately 0.15 to 0.5, reflecting variation in model capacity and compression settings. Gradient norms remained stable throughout training, indicating that optimization was not affected by vanishing gradients. Notably, we did not observe the exponentially small gradient magnitudes that are characteristic of barren plateaus, even in deeper circuits with up to 30 layers. Overall, the convergence pattern was highly consistent: every quantum autoencoder reached a stable loss in the early stages of training. These results confirm that the quantum autoencoder can be trained efficiently on large-scale datasets, enabling subsequent evaluation of learned representations on downstream peptide classification tasks.

### 4.2 CLASSIFICATION RESULTS

We evaluated the learned quantum autoencoder representations on seven peptide binary classification tasks and compared them with the Hamiltonian baseline. In parallel, we evaluated PLMs against classical baselines to provide a reference of state-of-the-art performance on these tasks. The results of these two comparisons are shown in Figure 4. Following the pre-training setup, classification performance was evaluated using the best-performing quantum autoencoder variant for each dataset, selected by cross-validation. The downstream evaluation procedure for quantum kernels is described in Section 3.4. Relative to the Hamiltonian kernel baseline, the quantum autoencoder improved classification accuracy across all datasets, with an average gain of +2.9%. Absolute mean ± std accuracies for each dataset are provided in Table 1. This indicates that unsupervised quantum autoencoder pre-training can capture non-trivial features beyond those present in the fixed Hamiltonian encoding. We further compare the pretrained quantum autoencoder against the same architecture trained directly on each downstream dataset. Pretraining consistently yields higher accuracy across all tasks, indicating that large-scale unsupervised learning provides transferable quantum representations. Full results are provided in Appendix H.

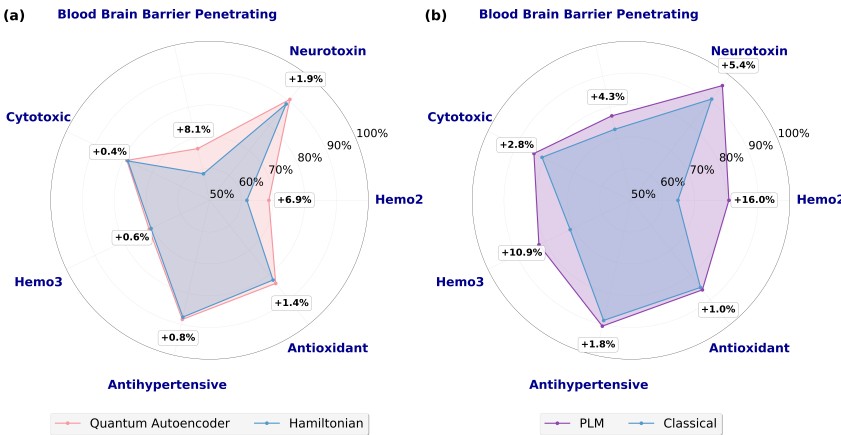

Figure 4: Test accuracy comparison across seven peptide datasets. (a) Quantum autoencoder representations compared to Hamiltonian baselines. (b) PLM representations compared to classical baselines. The two comparisons use different baselines and are therefore not directly comparable, but both highlight the benefit of representation learning. (Mean values used in this plot correspond to the entries reported in Table 1, where full mean ± std results are provided.)

Table 1: **Per-dataset comparison between the best-performing quantum autoencoder (QAE) configuration and the Hamiltonian baseline.** Values report mean ± std over 5-fold cross-validation with 20 random seeds. (QAE Config: L denotes the number of layers, and T denotes the number of traced-out qubits.)

| Dataset | # Qubit | QAE Config | Hamiltonian | QAE | Δ |
|---|---|---|---|---|---|
| Antihypertensive | 10 | 30L-1T | $0.8768_{\pm0.0036}$ | $\mathbf{0.8844_{\pm0.0024}}$ | $+0.0076$ |
| Antioxidant | 8 | 30L-2T | $0.8208_{\pm0.0004}$ | $\mathbf{0.8344_{\pm0.0079}}$ | $+0.0136$ |
| BBB | 8 | 30L-2T | $0.5856_{\pm0.0738}$ | $\mathbf{0.6667_{\pm0.0000}}$ | $+0.0810$ |
| Cytotoxic | 8 | 30L-1T | $0.7852_{\pm0.0000}$ | $\mathbf{0.7889_{\pm0.0000}}$ | $+0.0038$ |
| Hemo2 | 8 | 30L-2T | $0.6177_{\pm0.0653}$ | $\mathbf{0.6869_{\pm0.0000}}$ | $+0.0692$ |
| Hemo3 | 8 | 30L-1T | $0.7044_{\pm0.0000}$ | $\mathbf{0.7107_{\pm0.0000}}$ | $+0.0063$ |
| Neurotoxin | 10 | 30L-2T | $0.8871_{\pm0.0000}$ | $\mathbf{0.9058_{\pm0.0019}}$ | $+0.0187$ |

For the classical experiments, the baseline was chosen as the best of five candidate classifiers trained directly on flattened one-hot encoding, while the PLM results corresponds to the best of 12 pre-trained models. PLMs achieved larger improvements, with an average gain of +6.0%. The representation pipelines and evaluation procedure for these experiments are described in Section 3.1.

Although the two radar plots are not directly comparable due to different baselines, both highlight the importance of representation learning. Learned representations consistently outperformed fixed encodings in both quantum and classical settings. Together, these results establish the quantum autoencoder as a proof of principle of a quantum analogue of PLM-style pre-training.

At the same time, several limitations remain. While the quantum autoencoder improves performance across all datasets (0.4–8.1% over Hamiltonian baselines), the magnitude of these gains is smaller than those observed for PLMs(1–16% over classical baselines). All experiments were performed using noiseless quantum simulators, without accounting for hardware noise. Moreover, the Hamiltonian encoding may not be optimal for long peptide sequences, suggesting room for alternative encoding strategies. Another limitation is that quantum evaluations were restricted to kernel-based SVMs. Unlike PLMs or one-hot encodings, which output feature vectors that can be fed directly to diverse classifiers, the quantum autoencoder produces reduced density matrices as representations. Using these density matrices as feature vectors would require full state tomography (Vogel & Risken, 1989), which is prohibitively expensive for large systems (Haah et al., 2016). Although classical shadow provides an efficient alternative for predicting local properties of quantum states (Huang et al., 2020; Wu et al., 2023a; Nguyen et al., 2022; Akhtar et al., 2022; Bertoni et al., 2022), its

applicability to long-range entanglement properties still requires an exponential amount of samples. For this reason, quantum kernel methods currently provide the most straightforward way to benchmark quantum representations.

**Comparison with other fixed quantum encodings.** To ensure that the Hamiltonian encoding is not a weak baseline, we also evaluated two commonly used fixed encodings, angle encoding and amplitude encoding under the same qubit budget and downstream kernel-SVM classifier. Across all seven datasets, Hamiltonian evolution is the stronger fixed encoding overall, outperforming angle encoding on every dataset and amplitude encoding on five out of seven datasets. This confirms that the Hamiltonian baseline used throughout the paper is competitive rather than weak. Full results are provided in Appendix G.

**Classical representation-learning baselines.** For completeness, we also evaluated classical autoencoder (AE) and variational autoencoder (VAE) representations pretrained on the same 3 million peptide sequences. These models have parameter counts on the order of $10^3$, comparable to the quantum autoencder models used here (at most 900 trainable parameters). This comparison is intended only to provide context within broadly similar unsupervised representation learning approaches; we do not draw any claims or conclusions from this comparison, as a fair and systematic analysis of classical versus quantum representation learners would require an extensive study of their model structures, capacities, and optimization behavior. Detailed results, including best-classifier and SVM-only evaluations, are provided in Appendix I.

### 4.3 NOISE ROBUSTNESS

To assess whether the quantum autoencoder's improvements persist under realistic hardware noise, we evaluated both the Hamiltonian baseline and the best quantum autoencoder models under depolarizing noise with strengths $p \in \{0.00, 0.02, 0.05, 0.10, 0.20\}$. Across all datasets, learned representations remain stable under moderate noise and continue to outperform or match the Hamiltonian baseline at all noise levels. In several cases, the Hamiltonian baseline even improves slightly at higher $p$, consistent with mild regularization effects. A complete per-dataset table is provided in Appendix F.

## 5 CONCLUSION

We have presented a quantum autoencoder approach for representation learning on peptide sequences. Quantum autoencoders were shown to train efficiently on a large-scale peptide corpus without encountering barren plateaus, and consistently improved classification performance over Hamiltonian baselines across seven datasets. These results demonstrate that unsupervised quantum pre-training can extract non-trivial features from biological sequences, providing a quantum analogue of representation learning. While the empirical improvements achieved by the quantum autoencoder are moderate in magnitude, this is expected given both the strength of the Hamiltonian baseline and the highly compressed 8–10 qubit representations used throughout our evaluation. In this restricted setting, the aim is less about achieving large absolute accuracy gains and more about demonstrating that unsupervised quantum representation learning can capture data-driven structure from large corpora and transfer these features across downstream tasks. The results therefore provide an early but meaningful indication that learned quantum representations can complement fixed encodings within resource-limited quantum pipelines.

At the same time, the smaller magnitude of improvement, relative to the gains achieved by PLMs over classical baselines reflects the early stage of quantum representation learning and highlights the substantial room for progress as the field matures. In line with this, much of the existing QML literature still relies on hard-coded, task-agnostic encoding strategies for mapping classical data into quantum states. Such fixed encodings often fail to adapt to the statistical structure of real-world data, limiting their downstream effectiveness. Our results suggest that learned quantum representations, enabled by quantum autoencoder pre-training, offer a promising alternative. Potential future work includes scaling quantum autoencoders to larger architectures, developing hybrid quantum-classical pipelines, and integrating with PLMs to combine the strengths of both paradigms. Together, these directions have strong potential to advance QML in bioinformatics applications.

## REPRODUCIBILITY STATEMENT

All datasets and source code necessary to reproduce our results will be made publicly available upon acceptance through a GitHub repository. For review purposes, these materials are currently accessible via an anonymous repository: `https://anonymous.4open.science/r/quantum-autoencoder-bioinformatics-747D/README.md`, while the large pretraining dataset (3M peptides) is provided as supplementary material due to size constraints. Theoretical proofs of variational circuit depth complexity and sample complexity of the quantum autoencoder are provided in Appendix A and B respectively.

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

## A VARIATIONAL CIRCUIT DEPTH COMPLEXITY OF QUANTUM AUTOENCODER

Consider the initial quantum feature state $|\psi(\boldsymbol{x}_i)\rangle = e^{-iH(\boldsymbol{x}_i)t}|0^n\rangle$ with the local Hamiltonian matrix $H(\boldsymbol{x}_i) = \sum_{s=1}^{L} \boldsymbol{x}_i(s)P_s$, where $P_s$ represents a 'constant-weight' Pauli operator non-trivially acting on constant number of qubits. The essential idea of quantum autoencoder aims to train a quantum circuit $U(\vec{\boldsymbol{\theta}})$, enabling

$$U(\vec{\boldsymbol{\theta}})\left(\mathbb{E}_{\boldsymbol{x}_i \sim \mathcal{D}}\left[|\psi(\boldsymbol{x}_i)\rangle\langle\psi(\boldsymbol{x}_i)|\right]\right)U^\dagger(\vec{\boldsymbol{\theta}}) \approx |0^m\rangle\langle 0^m| \otimes \sigma, \tag{5}$$

where $\sigma$ represents a $(n-m)$-qubit density matrix, and the number of decoupled qubits $m$ is a constant. In the main file, we suggested that the quantum autoencoder circuit $U(\vec{\boldsymbol{\theta}})$ can be trained via minimizing a loss function. We first evaluate the necessary quantum circuit depth of $U(\vec{\boldsymbol{\theta}})$.

*Proof of Theorem 1 in the main text.* The fundamental idea within quantum autoencoder is to approximately decouple $m$ qubits from the whole quantum system. Without loss of generality, we assume the Hamiltonian $H(\boldsymbol{x}_i)$ is defined on a constant-dimensional lattice, and $\max\{\|\boldsymbol{x}_i\|_\infty |t|\}$ is a constant which does not increase with the system size. According to the Lieb-Robinson bound (Haah et al., 2021), the required quantum circuit depth for the quantum feature state $|\psi(\boldsymbol{x}_i)\rangle$ is $\mathrm{poly}\log(nt/\epsilon)$. On the other hand, we label the target qubits to be trashed as $Q = \{q_1, \cdots, q_m\}$. Since the information only spreads within the light-cone, as a result, a quantum circuit $U(\vec{\boldsymbol{\theta}})$ with circuit depth $d = \mathrm{poly}\log(nt/\epsilon)$ suffices to decouple qubit set $Q$ back to $|0^m\rangle$. $\qquad\square$

## B SAMPLE COMPLEXITY OF QUANTUM AUTOENCODER

Let the density matrix $\rho_\mathcal{D} = \mathbb{E}_{\boldsymbol{x}_i \sim \mathcal{D}}\left(|\psi(\boldsymbol{x}_i)\rangle\langle\psi(\boldsymbol{x}_i)|\right)$, then one can rewrite the loss function

$$\mathcal{L}(\vec{\boldsymbol{\theta}}) = \mathrm{Tr}\left[\Pi_m U(\vec{\boldsymbol{\theta}})\rho_\mathcal{D}U^\dagger(\vec{\boldsymbol{\theta}})\right] = \mathrm{Tr}\left[U^\dagger(\vec{\boldsymbol{\theta}})\Pi_m U(\vec{\boldsymbol{\theta}})\rho_\mathcal{D}\right]. \tag{6}$$

Here, we aim to answer the question "*How many samples to $\rho_\mathcal{D}$ suffices to estimate $\mathcal{L}(\vec{\boldsymbol{\theta}})$ within $\epsilon$ additive error for all candidate $\vec{\boldsymbol{\theta}}$*," by studying the support size of the operator $U^\dagger(\vec{\boldsymbol{\theta}})\Pi_m U(\vec{\boldsymbol{\theta}})$. We first give the main theoretical result, then detail the proof.

*Proof of Theorem 2 in the main text.* Suppose the variational quantum circuit

$$U(\vec{\boldsymbol{\theta}}) = U_d(\boldsymbol{\theta}_d)U(\boldsymbol{\theta}_{d-1})\cdots U_1(\boldsymbol{\theta}_1), \tag{7}$$

where each $U(\boldsymbol{\theta}_l)$ represents a layer of two-qubit gates $U_{j,l}$ for $l \in [d]$. Without loss of generality, we assume the distribution of each two-qubit gate $U_{j,l}$ is "locally scrambling", that is invariant under single-qubit rotations. This property is satisfied by a wide range of deep and shallow unstructured parameterised quantum circuits.

The proof is based on the backward propagation in the context of Heisenberg picture of the operator $O_{U(\vec{\boldsymbol{\theta}})} = U^\dagger(\vec{\boldsymbol{\theta}})\Pi_m U(\vec{\boldsymbol{\theta}})$. For the last layer of $U(\vec{\boldsymbol{\theta}})$, since $|\mathrm{supp}(\Pi_m)| = \mathcal{O}(1)$, and $U_d(\boldsymbol{\theta}_1)$ is a layer of two-qubit gates, it is straightforward to show that

$$O_d = U_d^\dagger(\boldsymbol{\theta}_1)\Pi_m U_d(\boldsymbol{\theta}_1) = \sum_{P \in \{I,X,Y,Z\}^{\otimes n}, \, |P| \leq m} \mathrm{Tr}[PU_d^\dagger(\boldsymbol{\theta}_1)\Pi_m U_d(\boldsymbol{\theta}_1)]P/2^n. \tag{8}$$

Then for index $l \in \{d-1, d-2, \cdots, 1\}$, let the operator

$$O_j = \frac{1}{2^n} \sum_{P \in \{I,X,Y,Z\}^{\otimes n}, \, |P| \leq k} \mathrm{Tr}[U_j^\dagger(\boldsymbol{\theta}_j)O_{j-1}U_j(\boldsymbol{\theta}_j)P]P, \tag{9}$$

until the final truncated observable

$$O^{(k)}_{U(\vec{\boldsymbol{\theta}})} = U_1^\dagger(\boldsymbol{\theta}_1)O_1 U_1(\boldsymbol{\theta}_1). \tag{10}$$

According to the Lemma 11 in the Ref. (Angrisani et al., 2024), it is shown that

$$\mathbb{E}_{U(\vec{\theta})}\left[\left|\mathrm{Tr}\left[\left(O_{U(\vec{\theta})} - O_{U(\vec{\theta})}^{(k)}\right)\rho\right]\right|\right] \leq \left(\frac{2}{3}\right)^{(k+1)/2} \|\Pi_m\|_{\mathrm{Pauli},2} \tag{11}$$

for any valid quantum state $\rho$, where the Pauli-2 norm is defined as

$$\|\Pi_m\|_{\mathrm{Pauli},2} = \left(\sum_P |\mathrm{Tr}[\Pi_m P]|^2\right)^{1/2}. \tag{12}$$

In our case, the operator $\Pi_m$ non-trivially acts on constant number of qubits, resulting in $\|\Pi_m\|_{\mathrm{Pauli},2} = \mathcal{O}(1)$. Let $k = \mathcal{O}(\log(1/\epsilon\delta))$, the above result immediately yields

$$\left|\mathrm{Tr}\left[\left(O_{U(\vec{\theta})} - O_{U(\vec{\theta})}^{(k)}\right)\rho\right]\right| \leq \epsilon \tag{13}$$

for any variational quantum circuit $U(\vec{\theta})$ except for a small fraction $\delta$.

Using this observation, we can further rewrite the quantum autoencoder loss function as

$$\mathcal{L}_{\mathrm{QAE}}(\vec{\theta}) = \mathrm{Tr}\left[O_{U(\vec{\theta})}^{(k)}\rho_{\mathcal{D}}\right], \tag{14}$$

where the operator $O_{U(\vec{\theta})}^{(k)} = \sum_{|P|\leq k} \alpha_P P$, which contains $M = \mathcal{O}(n^k)$ local Pauli terms. By leveraging the classical shadow method (Huang et al., 2020), it is shown that $N = \mathcal{O}\left(3^{\max_P\{|\mathrm{supp}(P)|\}} \log(M)\epsilon^{-2}\right)$ samples suffice to estimate $\mathcal{L}_{\mathrm{QAE}}(\vec{\theta})$. This completes the proof. $\qquad\square$

## C GENERALIZATION PERSPECTIVE ON QUANTUM AUTOENCODER IMPROVED KERNELS

Recall that the original quantum kernel function

$$k(\boldsymbol{x}_i, \boldsymbol{x}_j) = |\langle\phi(\boldsymbol{x}_i)|\phi(\boldsymbol{x}_j)\rangle|^2 = 1 - \mathrm{dist}_{\mathrm{tr}}^2(|\phi(\boldsymbol{x}_i)\rangle\langle\phi(\boldsymbol{x}_i)|, |\phi(\boldsymbol{x}_j)\rangle\langle\phi(\boldsymbol{x}_j)|),$$

where $\mathrm{dist}_{tr}(\cdot, \cdot)$ represents the trace distance. As a CPTP map, the quantum autoencoder (QAE) approach enables the relationship

$$\mathrm{dist}_{\mathrm{tr}}\left(\phi(\boldsymbol{x}_i), \phi(\boldsymbol{x}_j)\right) \leq \mathrm{dist}_{\mathrm{tr}}(|\phi(\boldsymbol{x}_i)\rangle\langle\phi(\boldsymbol{x}_i)|, |\phi(\boldsymbol{x}_j)\rangle\langle\phi(\boldsymbol{x}_j)|), \tag{15}$$

where the quantum feature state $\phi(\boldsymbol{x}_i) = \mathrm{QAE}\left(|\phi(\boldsymbol{x}_i)\rangle\langle\phi(\boldsymbol{x}_i)|\right)$. Equivalently, the related quantum kernel function satisfies

$$k_{\mathrm{QAE}}(\boldsymbol{x}_i, \boldsymbol{x}_j) \geq k(\boldsymbol{x}_i, \boldsymbol{x}_j) \tag{16}$$

for each data pair $(i, j)$.

In the context of the SVM model, the optimal primal value equals the optimal dual value, that is

$$p_K = \min_{w,b} \frac{1}{2}\|w\|^2 = \max_\alpha W_K(\alpha), \tag{17}$$

where the dual SVM objective for the kernel $K$ is defined by

$$W_K(\alpha) = \sum_i \alpha_i - \frac{1}{2}\sum_{i,j} \alpha_i \alpha_j y_i y_j K_{ij}. \tag{18}$$

Fix any parameter $\alpha$, we consider the difference

$$W_{K_{\mathrm{QAE}}}(\alpha) - W_K(\alpha) = -\frac{1}{2}\sum \alpha_i \alpha_j y_i y_j \Delta_{ij}, \tag{19}$$

where $\Delta_{ij} = [K_{\mathrm{QAE}}]_{ij} - [K]_{ij} \geq 0$. This further results in

$$W_{K_{\mathrm{QAE}}}(\alpha) - W_K(\alpha) \leq 0, \tag{20}$$

which directly yields $p_{K_{\mathrm{QAE}}} \leq p_K$. A standard margin-based generalization upper bound states that with probability $\geq 1 - \delta$ over an i.i.d. sample of size $N$, every linear classifier in the RKHS with margin $1/p_K$ and feature space radius $R$ satisfies

$$\epsilon_{\mathrm{gen}} = \mathcal{O}(R^2 p_K/N) + \mathcal{O}(\sqrt{\log(1/\delta)/N}). \tag{21}$$

This indicates that $\epsilon_{\mathrm{gen}}(K_{\mathrm{QAE}}) \leq \epsilon_{\mathrm{gen}}(K)$, implying QAE method may increase the generalization ability for quantum kernel method.

## D  PROTEIN LANGUAGE MODELS USED

Table 2 lists all PLMs considered in this work, along with the dimensionalities of their representations.

Table 2: List of pre-trained PLMs used in this study.

| Model | Representation Dimension |
|---|---|
| ESM-2 (t6_8M) (Lin et al., 2023) | 320 |
| ESM-2 (t12_35M) (Lin et al., 2023) | 480 |
| ESM-2 (t30_150M) (Lin et al., 2023) | 640 |
| PeptideBERT (Guntuboina et al., 2023) | 768 |
| BioBERT (Lee et al., 2020) | 768 |
| Ankh-base (Elnaggar et al., 2023) | 768 |
| ProtBERT Elnaggar et al. (2021) | 1024 |
| ProtBERT-BFD Elnaggar et al. (2021) | 1024 |
| ProtT5-XL Elnaggar et al. (2021) | 1024 |
| ESM-2 (t33_650M) (Lin et al., 2023) | 1280 |
| ESM-1b (Rives et al., 2021) | 1280 |
| Ankh-large (Elnaggar et al., 2023) | 1536 |

## E  DATASETS

### E.1  PRE-TRAINING DATASET

**UniProt Peptide Dataset.**  We derived our dataset from the UniProt Knowledgebase (UniProtKB) (uni, 2025), which provides a comprehensive, curated, and regularly updated repository of protein sequences from diverse species. Using the UniProt search interface, we extracted 3,062,374 unique peptide sequences with lengths between 4 and 50 amino acids. Both Swiss-Prot (reviewed) and TrEMBL (unreviewed) entries were included (Bairoch & Apweiler, 2000; Jungo et al., 2012), ensuring coverage of a wide range of species, biological functions, and environmental contexts. The query was performed on UniProt [14 March 2024], applying sequence-length filters to restrict results to the peptidome. Redundant entries were removed, and only sequences within the defined length window were retained. This dataset thus provides a broad and representative sample of the global peptidome, which we used as the basis for unsupervised training of our quantum autoencoder models.

### E.2  CLASSIFICATION DATASETS

Table 3: Number of sequences for each dataset, showing positive and negative counts in the train and test splits after preprocessing.

| Dataset | #Sequences | Train (Pos/Neg) | Test (Pos/Neg) |
|---|---|---|---|
| Antihypertensive | 2,778 | 1,135 / 809 | 487 / 347 |
| Antioxidant | 1,933 | 745 / 608 | 319 / 261 |
| Blood-Brain Barrier-Penetrating | 650 | 248 / 207 | 107 / 88 |
| Cytotoxic | 2,651 | 1,017 / 838 | 436 / 360 |
| Hemo 2 | 992 | 435 / 359 | 108 / 90 |
| Hemo 3 | 1,587 | 692 / 577 | 173 / 145 |
| Neurotoxin | 1033 | 344 / 379 | 147 / 163 |

**Data collection process.**  All datasets were collected from the Peptipedia v2.0 database using its query system for biological activities(Cabas-Mora et al., 2024). Peptides were selected based on the desired function and further filtered to exclude sequences containing non-canonical residues or chemical modifications. Additional constraints were applied to retain only experimentally validated

peptides with a maximum length of 100 residues. For the generation of negative examples, the Peptipedia v2.0 database was also used to select peptides lacking the target activity, applying the same filtering criteria to ensure consistency across datasets(Cabas-Mora et al., 2024).

**Dataset preparation.** To prepare the datasets for the training process, a homology reduction step was performed using CD-Hit [14]. Each dataset was first divided into positive and negative examples. For both subsets, CD-Hit was applied with a 90% sequence identity threshold to reduce redundancy. Subsequently, undersampling strategies were implemented to balance the datasets by adjusting the number of negative examples to match the positive ones (Lin et al., 2017). To achieve this, sequences with reduced homology were randomly selected from the clusters generated by CD-Hit, ensuring a balanced representation across clusters.

**Preprocessing.** For each downstream dataset, we examined sequence length distributions using histograms. We observed that beyond a certain length threshold, all sequences were sparsely represented, so sequences longer than this threshold were removed (e.g., sequences longer than 65 amino acids in one dataset). The threshold varied across datasets and was chosen manually. After filtering, sequences were one-hot encoded, shorter sequences were padded to match the maximum sequence length within the dataset, and the resulting matrices were flattened into binary vectors for input to the models. Table 3 reports the number of sequences in each dataset, along with the positive and negative counts in the train and test splits.

**Antihypertensive.** Antihypertensive peptides are short amino acid sequences that contribute to lowering blood pressure by modulating key biological and cellular processes involved in cardiovascular regulation (Liang et al., 2025). Their activity is often linked to the inhibition of enzymes such as angiotensin-converting enzyme (ACE) or the modulation of signaling pathways that influence vascular tone and fluid balance [11]. Beyond direct blood pressure control, these peptides may also mitigate hypertension-associated inflammation and immune responses, thereby reducing the risk of long-term cardiovascular damage (Ichim et al., 2024).

**Antioxidant.** Antioxidant peptides are short amino acid sequences with inherent bioactivity that protect cells and tissues from oxidative damage by counteracting reactive oxygen species (ROS) and other prooxidants (Zou et al., 2016). Their activity arises from multiple mechanisms, including direct radical scavenging, inhibition of lipid peroxidation, and chelation of transition metals that catalyze oxidative reactions (Zhang et al., 2022). By mitigating oxidative stress, these peptides contribute to the preservation of cellular integrity and metabolic balance, which is essential for overall health and the prevention of chronic diseases linked to oxidative damage (Zhu et al., 2024).

**Blood-brain barrier-penetrating.** Blood–brain barrier–penetrating peptides (BBBPs) are short amino acid sequences specifically designed or discovered to overcome the highly selective nature of the blood–brain barrier (BBB), a physiological shield that protects the central nervous system while severely limiting drug entry (Kumar et al., 2021). By interacting with receptors or transport systems on the endothelial cells that form the BBB, these peptides act as molecular shuttles that enable therapeutic or diagnostic molecules to reach the brain (Gu et al., 2024). Their sequences often feature amphipathic or positively charged residues that promote membrane interactions, and they can be chemically optimized to resist enzymatic degradation in circulation (Kumar et al., 2021).

**Cytotoxic.** Cytotoxic peptides are short amino acid sequences that induce cell death by directly interacting with and disrupting cell membranes, compromising their structural integrity (Luan et al., 2021). This membrane-targeting mechanism allows them to bypass intracellular resistance pathways and makes them particularly effective against rapidly dividing or abnormal cells, such as cancer cells (Luan et al., 2021). Because of their potent ability to trigger apoptosis or necrosis, cytotoxic peptides have been widely investigated as therapeutic agents in oncology, where they offer a promising alternative or complement to conventional chemotherapies (Rafieezadeh & Esfandyari, 2024).

**Hemo 2 & Hemo 3.** HemoPI-2 and HemoPI-3 datasets are compilations of experimentally validated peptides labeled as hemolytic or non-hemolytic which have been extracted from various sources, including the Hemolytik database (Gautam et al., 2014), Swiss-Prot (Jungo et al., 2012)

(a curated protein sequence database which is part of UniProt (uni, 2025)) and the Database of Antimicrobial Activity and Structure of Peptides (DBAASP) (Pirtskhalava et al., 2016). Hemolytic peptides are short amino acid sequences that have the ability to disrupt red blood cell membranes and cause cell lysis, whereas non-hemolytic peptides lack such activity and are considered safer for therapeutic applications.

**Neurotoxin.** A neurotoxin peptide is a short amino acid sequence that exerts toxic effects on the nervous system by disrupting neuronal communication and function. These peptides typically act by binding to and modulating ion channels, receptors, or neurotransmitter release machinery, leading to altered excitability, impaired synaptic transmission, or cell death (Zhou et al., 2025). Many are naturally produced by venomous organisms such as cone snails, scorpions, or spiders as a means of predation or defense, where their high potency and specificity allow them to immobilize prey or deter predators (Lee et al., 2021). While inherently harmful, neurotoxin peptides are also valuable molecular tools for probing the physiology of neuronal signaling and are being explored as scaffolds for therapeutic applications, where their precision in targeting neural pathways can be redirected for beneficial outcomes (Yadav et al., 2021).

## F    NOISE ROBUSTNESS RESULTS

**Implementation of depolarizing noise.** To efficiently evaluate noise robustness across all datasets, we applied depolarizing noise directly to the trace distance kernel entries rather than simulating noise inside the circuit. For depolarizing noise of strength $p$, the fidelity between any two quantum states is known to transform as

$$k_{ij}^{(p)} = (1 - p)\, k_{ij} + p \cdot \tfrac{1}{2},$$

which corresponds to each state being mixed with the maximally mixed state. This closed form update is the exact action of the single-qubit depolarizing channel on fidelity-based kernels, and therefore provides the same qualitative effect as circuit level depolarizing noise while being substantially more efficient to compute.

Table 4: **Noise robustness under depolarizing noise for Hamiltonian and QAE models.** Values report mean test accuracy (5-fold CV, 20 seeds) for noise levels $p \in \{0.00, 0.02, 0.05, 0.10, 0.20\}$.

| Dataset | Method | p=0.00 | p=0.02 | p=0.05 | p=0.10 | p=0.20 |
|---------|--------|--------|--------|--------|--------|--------|
| Antihypertensive | QAE | 0.8844 | 0.8825 | 0.8825 | 0.8832 | 0.8799 |
| | Hamiltonian | 0.8768 | 0.8773 | 0.8789 | 0.8773 | 0.8765 |
| Antioxidant | QAE | 0.8344 | 0.8360 | 0.8379 | 0.8345 | 0.8345 |
| | Hamiltonian | 0.8208 | 0.8224 | 0.8241 | 0.8241 | 0.8214 |
| BBB | QAE | 0.6667 | 0.6667 | 0.6615 | 0.6564 | 0.6462 |
| | Hamiltonian | 0.5856 | 0.6010 | 0.6226 | 0.6800 | 0.6826 |
| Cytotoxic | QAE | 0.7889 | 0.7902 | 0.7902 | 0.7877 | 0.7876 |
| | Hamiltonian | 0.7852 | 0.7839 | 0.7839 | 0.7839 | 0.7827 |
| Hemo2 | QAE | 0.6869 | 0.6818 | 0.6818 | 0.6566 | 0.6162 |
| | Hamiltonian | 0.6177 | 0.6177 | 0.6308 | 0.6308 | 0.6465 |
| Hemo3 | QAE | 0.7107 | 0.7107 | 0.7107 | 0.7013 | 0.6824 |
| | Hamiltonian | 0.7044 | 0.7044 | 0.7075 | 0.7013 | 0.6887 |
| Neurotoxin | QAE | 0.9058 | 0.9058 | 0.9045 | 0.9055 | 0.9018 |
| | Hamiltonian | 0.8871 | 0.8887 | 0.8871 | 0.8871 | 0.8913 |

# G   COMPARISON WITH OTHER FIXED QUANTUM ENCODINGS

Table 5: **Performance of fixed quantum encodings (Hamiltonian, Angle, Amplitude) compared with QAE.** Values report mean ± std over 5-fold CV with 20 seeds.

| Dataset | Hamiltonian | Angle | Amplitude | QAE |
|---|---|---|---|---|
| Antihypertensive | $0.8768_{\pm 0.0036}$ | $0.7902_{\pm 0.0000}$ | $\mathbf{0.8969_{\pm 0.0000}}$ | $0.8844_{\pm 0.0024}$ |
| Antioxidant | $0.8208_{\pm 0.0004}$ | $0.7017_{\pm 0.0759}$ | $0.5500_{\pm 0.0000}$ | $\mathbf{0.8344_{\pm 0.0079}}$ |
| BBB | $0.5856_{\pm 0.0738}$ | $0.5487_{\pm 0.0000}$ | $0.5487_{\pm 0.0000}$ | $\mathbf{0.6667_{\pm 0.0000}}$ |
| Cytotoxic | $0.7852_{\pm 0.0000}$ | $0.5477_{\pm 0.0000}$ | $0.5690_{\pm 0.0156}$ | $\mathbf{0.7889_{\pm 0.0000}}$ |
| Hemo2 | $0.6177_{\pm 0.0653}$ | $0.5455_{\pm 0.0000}$ | $0.5455_{\pm 0.0000}$ | $\mathbf{0.6869_{\pm 0.0000}}$ |
| Hemo3 | $0.7044_{\pm 0.0000}$ | $0.5440_{\pm 0.0000}$ | $0.6069_{\pm 0.0000}$ | $\mathbf{0.7107_{\pm 0.0000}}$ |
| Neurotoxin | $0.8871_{\pm 0.0000}$ | $0.5258_{\pm 0.0000}$ | $\mathbf{0.9410_{\pm 0.0015}}$ | $0.9058_{\pm 0.0019}$ |

# H   VALUE OF PRETRAINING: QAE VS. TRAINING FROM SCRATCH

Table 6: **Comparison of pretrained QAEs and QAEs trained from scratch on each downstream dataset.** Values are mean ± std over 5-fold CV with 20 random seeds.

| Dataset | Hamiltonian | QAE (Pretrained) | QAE (Downstream trained) |
|---|---|---|---|
| Antihypertensive | $0.8768_{\pm 0.0036}$ | $\mathbf{0.8844_{\pm 0.0024}}$ | $0.8813_{\pm 0.0000}$ |
| Antioxidant | $0.8208_{\pm 0.0004}$ | $\mathbf{0.8344_{\pm 0.0079}}$ | $0.7955_{\pm 0.0025}$ |
| BBB | $0.5856_{\pm 0.0738}$ | $\mathbf{0.6667_{\pm 0.0000}}$ | $0.6308_{\pm 0.0000}$ |
| Cytotoxic | $0.7852_{\pm 0.0000}$ | $\mathbf{0.7889_{\pm 0.0000}}$ | $0.7764_{\pm 0.0000}$ |
| Hemo2 | $0.6177_{\pm 0.0653}$ | $\mathbf{0.6869_{\pm 0.0000}}$ | $0.6558_{\pm 0.0253}$ |
| Hemo3 | $0.7044_{\pm 0.0000}$ | $\mathbf{0.7107_{\pm 0.0000}}$ | $0.7013_{\pm 0.0000}$ |
| Neurotoxin | $0.8871_{\pm 0.0000}$ | $\mathbf{0.9058_{\pm 0.0019}}$ | $0.8890_{\pm 0.0039}$ |

# I   CLASSICAL AUTOENCODER AND VAE BASELINES

Table 7: **Comparison of QAE and classical learned representations (AE, VAE) using the best-performing classical classifier.** Values are mean accuracy over 5-fold CV with 20 seeds.

| Dataset | QAE | AE | VAE |
|---|---|---|---|
| Antihypertensive | $\mathbf{0.8844_{\pm 0.0024}}$ | $0.8370_{\pm 0.0018}$ | $0.8789_{\pm 0.0000}$ |
| Antioxidant | $\mathbf{0.8344_{\pm 0.0079}}$ | $0.7940_{\pm 0.0046}$ | $0.8063_{\pm 0.0075}$ |
| BBB | $0.6667_{\pm 0.0000}$ | $\mathbf{0.7026_{\pm 0.0000}}$ | $0.6462_{\pm 0.0000}$ |
| Cytotoxic | $\mathbf{0.7889_{\pm 0.0000}}$ | $0.7023_{\pm 0.0000}$ | $0.7148_{\pm 0.0000}$ |
| Hemo2 | $0.6869_{\pm 0.0000}$ | $\mathbf{0.6970_{\pm 0.0000}}$ | $0.6414_{\pm 0.0000}$ |
| Hemo3 | $\mathbf{0.7107_{\pm 0.0000}}$ | $0.6824_{\pm 0.0000}$ | $0.6698_{\pm 0.0000}$ |
| Neurotoxin | $\mathbf{0.9058_{\pm 0.0019}}$ | $0.8269_{\pm 0.0049}$ | $0.7968_{\pm 0.0000}$ |

Table 8: **Comparison of QAE and classical learned representations (AE, VAE) using only an SVM classifier.** Values are mean accuracy over 5-fold CV with 20 seeds.

| Dataset | QAE | AE | VAE |
|---|---|---|---|
| Antihypertensive | $0.8844_{\pm 0.0024}$ | $0.8056_{\pm 0.0005}$ | $0.8789_{\pm 0.0000}$ |
| Antioxidant | $0.8344_{\pm 0.0079}$ | $0.7940_{\pm 0.0046}$ | $0.6771_{\pm 0.0008}$ |
| BBB | $0.6667_{\pm 0.0000}$ | $0.6615_{\pm 0.0000}$ | $0.6218_{\pm 0.0168}$ |
| Cytotoxic | $0.7889_{\pm 0.0000}$ | $0.6972_{\pm 0.0000}$ | $0.6300_{\pm 0.0033}$ |
| Hemo2 | $0.6869_{\pm 0.0000}$ | $0.6616_{\pm 0.0000}$ | $0.5616_{\pm 0.0083}$ |
| Hemo3 | $0.7107_{\pm 0.0000}$ | $0.6547_{\pm 0.0019}$ | $0.6352_{\pm 0.0000}$ |
| Neurotoxin | $0.9058_{\pm 0.0019}$ | $0.8048_{\pm 0.0116}$ | $0.7348_{\pm 0.0135}$ |

## J   LARGE LANGUAGE MODELS USAGE STATEMENT

Large Language Models (LLMs) were used in this work for language polishing, grammar checking, and code improvement. Specifically, LLMs assisted with manuscript editing to improve clarity and grammatical accuracy. For coding tasks, the authors provided the primary code structure and logic, while LLMs were used to improve code efficiency, check for bugs, ensure consistency, and optimize implementation details. All core research ideas, methodologies, and experimental design are entirely the work of the authors.

