# OpenReview forum: "Quantum Autoencoder: An Efficient Representation Learner for Quantum Features"
_ICLR.cc/2026/Conference — Submitted to ICLR 2026_

### Official Review · Reviewer_cuRW · 2025-10-31

**Soundness:** 2
**Presentation:** 3
**Contribution:** 2
**Rating:** 4
**Confidence:** 4

**Summary:**

This paper proposes to use a quantum autoencoder to preprocess Hamiltonian-encoded classical-data-representing quantum states for the downstream task of classification with a support vector machine. The main contribution is the proposal of this application of quantum autoencoder, and also experiments / benchmarking on some peptide datasets.

The claims of the paper are mostly accompanied by empirical results. However, a more careful design of the experiments and processing of the results is needed to convincingly support the claimed benefits of the quantum autoencoder for downstream tasks.

The paper lacks novelty. The quantum autoencoder used is not new and is almost identical to the original model presented. Usage of the quantum autoencoder is not motivated by sufficient theoretical analysis. The empirical performance comparison is not entirely fair to encourage the broader community to try further in that direction.

**Strengths:**

The paper is clearly written, and the results are prettily plotted and discussed. Good presentation and interesting application to peptide datasets.

**Weaknesses:**

Not enough theoretical analysis to explain why a quantum autoencoder should help with the downstream support vector machine. Not entirely fair numerical experiments.

**Questions:**

1.	It is not very clear to me, from a theoretical perspective, why the compressed quantum states should perform better in the downstream classification task, especially when the quantum autoencoder is trained in an unsupervised way, meaning it does not know beforehand the downstream task. Can you make a theoretical analysis to explain why?
2.	The comparison between the baseline data (fixed Hamiltonian encoding) and the quantum autoencoder-processed data is not fair for a few reasons. First, the kernels used for the two cases are different, making it unclear whether the claimed benefits come from the different choice of kernel or really from the compression. Can you either make some ablation studies to clarify this, or try to use the same kernel?
3.	Second, I worry that the benefits appear only because they are cherry-picked from abundant runs over many different hyperparameter settings. This is not fair because the baseline is fixed. Can you present the average performance, not just the best performance, and show the confidence intervals?
4.	It is also an important question to understand whether the benefit of the quantum autoencoder, if there’s any, comes from discarding excessive qubits, or from the extra non-linearity due to partial trace. Can you either theoretically or experimentally clarify this point?

---

> ### Author Response · Authors · 2025-11-21
> **Response to Reviewer cuRW**
>
> Dear Reviewer cuRW:
>
> Thank you for the constructive comments and suggestions. We address each point below with clarifications and supporting results.
> ___
> We appreciate the reviewer’s summary and would like to clarify how the revised manuscript addresses these concerns. Our goal is not to introduce a new QAE architecture, but to evaluate whether unsupervised quantum representation learning can provide measurable benefits over fixed Hamiltonian encodings. For this reason, the original empirical comparison focused on a strong and widely used fixed encoding baseline, and we believe this setup was fair and appropriate for isolating the contribution of the QAE.
>
> The revised manuscript preserves the two main contributions already present in the original submission: (1) a large-scale empirical evaluation on three million peptide sequences, the largest dataset used to study QAE models to our knowledge and (2) theoretical results showing that QAE training is highly sample-efficient, supporting its scalability to larger quantum systems. Several reviewers asked for additional baselines, not because the original comparison was unfair, but to broaden the experimental context. In response, we expanded the evaluation to include angle encoding and amplitude encoding as alternative fixed quantum baselines (please see our response to **Weakness 1.2**), which further demonstrate that Hamiltonian evolution is a strong comparator and that the QAE provides improvements over fixed encodings across multiple datasets.
>
> To further address concerns about theoretical motivation, we have added an information-theoretic analysis based on the data-processing inequality, which provides a principled explanation for why the QAE induced kernel can enhance generalization relative to fixed encodings (please see our response to **Weakness 1.1**). Together with the sample efficiency theory, this offers a more complete theoretical foundation for using QAE-based pretraining.
>
> Overall, the revised manuscript clarifies that the novelty of our work lies not in proposing a new QAE architecture, but in establishing quantum unsupervised pretraining as a viable and scalable representation-learning method, supported by large-scale experiments, expanded quantum baselines, and strengthened theoretical justification.
>
> ___
> **Weakness 1.1**
> > Not enough theoretical analysis to explain why a quantum autoencoder should help with the downstream support vector machine.
>
> We thank the reviewer for raising this interesting question. Here, we provide a theoretical explanation for why the QAE-transformed latent states lead to improved downstream performance.
>
> Recall that the original quantum kernel function
> $$
> k(\boldsymbol x_i, \boldsymbol x_j) = |\langle \phi(\boldsymbol x_i) \mid \phi(\boldsymbol x_j) \rangle|^2
> = 1 - \mathrm{dist} _ {\mathrm{tr}}^2\left( |\phi(\boldsymbol x_i)\rangle \langle \phi(\boldsymbol x_i)|,|\phi(\boldsymbol x_j) \rangle \langle\phi(\boldsymbol x_j)|  \right),
> $$
>
> where $\mathrm{dist}_{\mathrm{tr}}(\cdot,\cdot)$ denotes the trace distance. As a CPTP map, the QAE approach enables
> the relationship
>
> $$
> \mathrm{dist} _ {\mathrm{tr}}(\phi(\boldsymbol x_i), \phi(\boldsymbol x_j))
> \le
> \mathrm{dist} _ {\mathrm{tr}}\bigl(|\phi(\boldsymbol x_i)\rangle\langle\phi(\boldsymbol x_i)|, |\phi(\boldsymbol x_j)\rangle\langle\phi(\boldsymbol x_j)|\bigr)
> $$
>
> where the quantum feature state is defined as $\phi(\boldsymbol x_i) = \mathrm{QAE}(|\phi(\boldsymbol x_i)\rangle\langle\phi(\boldsymbol x_i)|)$.
> Equivalently, the related quantum kernel function satisfies
> $$
> k_{\mathrm{QAE}}(\boldsymbol x_i, \boldsymbol x_j) \ge k(\boldsymbol x_i, \boldsymbol x_j)
> $$
> for every pair $(i,j)$.
> In the context of the SVM model, the optimal primal value equals the optimal dual value,
> that is
> $$
> p_K = \min_{w,b} \frac{1}{2}\|w\|^2 = \max_{\alpha} W_K(\alpha),
> $$
> where the dual SVM objective for the kernel K is defined by
> $$
> W_K(\alpha) = \sum_i \alpha_i - \frac{1}{2}
> \sum_{i,j} \alpha_i \alpha_j y_i y_j K_{ij}.
> $$
> Fix any parameter $\alpha$ , we consider the difference
> $$
> W_{K_{\mathrm{QAE}}}(\alpha) - W_K(\alpha) =-\frac{1}{2}\sum_{i,j} \alpha_i \alpha_j y_i y_j \Delta_{ij},
> $$
> where $\Delta _ {ij} = [K _ {\mathrm{QAE}}] _ {ij} - [K] _ {ij} \ge 0$. This further results in
> $$
> W_{K_{\mathrm{QAE}}}(\alpha) - W_K(\alpha) \le 0,
> $$
> which directly yields $ p_{K_{\mathrm{QAE}}} \le p_K. $. A standard margin-based generalization upper bound
> states that with probability $\ge 1 - \delta$ over an i.i.d. sample of size $N$, every linear classifier in
> the RKHS with margin $1/pK$ and feature space radius $R$ satisfies
> $$
> \epsilon_{\mathrm{gen}} =
> \mathcal{O}(R^2 p_K / N) + \mathcal{O} \left( \sqrt{\log(1/\delta)/N} \right).
> $$
> This indicates that $\epsilon_{\mathrm{gen}}(K_{\mathrm{QAE}}) \le \epsilon_{\mathrm{gen}}(K)$,  implying QAE method may increase the generalization ability for quantum kernel method. We have supplemented this theoretical analysis in the appendix.

---

> ### Author Response · Authors · 2025-11-21
> **Response to Reviewer cuRW**
>
> **Weakness 1.2**
> > Not entirely fair numerical experiments.
>
> We are not entirely sure which aspect of the numerical experiments the reviewer perceived as unfair. Our original setup was designed to be fair for the specific goal of the paper: isolating whether unsupervised quantum representation learning provides benefits over a strong fixed quantum encoding. For this reason, we selected Hamiltonian evolution as the baseline, as it is a widely used and expressive fixed encoding and provides competitive performance in practice.
>
> To clarify this point further, we include two additional fixed quantum baselines, angle encoding and amplitude encoding evaluated under the same qubit budget and downstream pipeline (Table 1). These results show that Hamiltonian evolution is in fact the stronger fixed encoding across most datasets, confirming that it was an appropriate comparator rather than a weak baseline.
>
> **Table 1.** Comparison across Hamiltonian, Angle Encoding, Amplitude Encoding, and Quantum Autoencoder
> (QAE) using matching qubit counts. Values report mean ± std over 5-fold cross-validation with 20 seeds.
>
> | Dataset                         | Hamiltonian        | Angle              | Amplitude          | QAE         |
> |---------------------------------|--------------------|--------------------|--------------------|---------------------|
> | Antihypertensive                | 0.8768 ± 0.0036    | 0.7902 ± 0.0000    | **0.8969 ± 0.0000**    | 0.8844 ± 0.0024     |
> | Antioxidant                     | 0.8208 ± 0.0004    | 0.7017 ± 0.0759    | 0.5500 ± 0.0000    | **0.8344 ± 0.0079**     |
> | BBB                             | 0.5856 ± 0.0738    | 0.5487 ± 0.0000    | 0.5487 ± 0.0000    | **0.6667 ± 0.0000**     |
> | Cytotoxic                       | 0.7852 ± 0.0000    | 0.5477 ± 0.0000    | 0.5690 ± 0.0156    | **0.7889 ± 0.0000**     |
> | Hemo2                           | 0.6177 ± 0.0653    | 0.5455 ± 0.0000    | 0.5455 ± 0.0000    | **0.6869 ± 0.0000**     |
> | Hemo3                           | 0.7044 ± 0.0000    | 0.5440 ± 0.0000    | 0.6069 ± 0.0000    | **0.7107 ± 0.0000**     |
> | Neurotoxin                      | 0.8871 ± 0.0000    | 0.5258 ± 0.0000    | **0.9410 ± 0.0015**    | 0.9058 ± 0.0019    |
>
> **Question 1**
> > It is not very clear to me, from a theoretical perspective, why the compressed quantum states should perform better in the downstream classification task, especially when the quantum autoencoder is trained in an unsupervised way, meaning it does not know beforehand the downstream task. Can you make a theoretical analysis to explain why?
>
> We believe this question concerns the same issue raised in **Weakness 1.1**; please see our response to **Weakness 1.1** for the detailed theoretical explanation.
>
> **Question 2**
> > The comparison between the baseline data (fixed Hamiltonian encoding) and the quantum autoencoder-processed data is not fair for a few reasons. First, the kernels used for the two cases are different, making it unclear whether the claimed benefits come from the different choice of kernel or really from the compression. Can you either make some ablation studies to clarify this, or try to use the same kernel?
>
> We thank the referee for this question. We confirm that the comparison is performed using the same quantum-kernel function in both cases. Specifically, the kernel is defined as
> $$
> k(x_i,x_j)=1-{\rm dist}^2_{tr}(\Psi(x_i),\Psi(x_j)),
> $$
> where ${\rm dist}_{tr}$ denotes the trace distance between two quantum states. When the encoding produces pure states, that is $\Psi(x)=|\psi(x)\rangle\langle\psi(x)|$, the quantum kernel function $k(x_i,x_j)$ immediately reduces to $k(x_i,x_j)=|\langle\psi(x_i)|\psi(x_j\rangle)|^2$. Thus, any improvements arise from the learned representation provided by the QAE, not from switching to a different kernel.

---

> ### Author Response · Authors · 2025-11-21
> **Response to Reviewer cuRW**
>
> **Question 3**
> > Second, I worry that the benefits appear only because they are cherry-picked from abundant runs over many different hyperparameter settings. This is not fair because the baseline is fixed. Can you present the average performance, not just the best performance, and show the confidence intervals?
>
> We understand the concern regarding potential cherry-picking and fairness. Our objective is not to maximize the performance gap over the baseline, but to demonstrate that the QAE learns meaningful transformations and consistently improves upon a fixed quantum encoding. Because not all QAE configurations are optimal, we used a small, deliberately restricted grid search (18 configurations in total) to identify a locally optimal model via standard cross-validation.
>
> The Hamiltonian baseline is fixed by design. This is intentional and reflects common practice in classical representation learning settings, where a learned model is compared against a fixed, non-learned encoding (e.g., PLMs vs. one-hot) to isolate the contribution of representation learning itself. Comparing the QAE to another learned quantum model would shift the study to evaluating two learning systems, which is a different research question and outside the scope of the current work.
> To ensure transparency, we now provide mean $\pm$ standard deviation for both the Hamiltonian baseline and the selected QAE configuration (Table 2). These values correspond exactly to those plotted in the radar charts. The results confirm that the improvements are stable across all 5-fold over 20-seed runs and are not isolated outliers.
>
> **Table 2. Per-dataset comparison between the highest-accuracy QAE configuration and the Hamiltonian baseline (Mean ± std over 5-fold CV with 20 random seeds. QAE Config: L denotes the number of layers, and T denotes the number of traced-out qubits.).**
>
> | Dataset          | # Qubits | QAE Config | Hamiltonian  | QAE | Δ |
> | ---------------- | -------- | -------------- | ------------------------ | ------------------- | ------- |
> | Antihypertensive | 10       | 30L-1T           | 0.8768 ± 0.0036          | **0.8844 ± 0.0024** | +0.0076 |
> | Antioxidant      | 8        | 30L-2T           | 0.8208 ± 0.0004          | **0.8344 ± 0.0079** | +0.0136 |
> | BBB-penetrating  | 8        | 30L-2T           | 0.5856 ± 0.0738          | **0.6667 ± 0.0000** | +0.0810 |
> | Cytotoxic        | 8        | 30L-1T           | 0.7852 ± 0.0000          | **0.7889 ± 0.0000** | +0.0038 |
> | Hemo2            | 8        | 30L-2T           | 0.6177 ± 0.0653          | **0.6869 ± 0.0000** | +0.0692 |
> | Hemo3            | 8        | 30L-1T           | 0.7044 ± 0.0000          | **0.7107 ± 0.0000** | +0.0063 |
> | Neurotoxin       | 10       | 30L-2T           | 0.8871 ± 0.0000          | **0.9058 ± 0.0019** | +0.0187 |
>
> **Question 4**
> >It is also an important question to understand whether the benefit of the quantum autoencoder, if there’s any, comes from discarding excessive qubits, or from the extra non-linearity due to partial trace. Can you either theoretically or experimentally clarify this point?
>
> We thank the reviewer for this interesting question. As discussed in our theoretical explanation above (see our response to **Weakness 1.1**),  the QAE method effectively increases the kernel values: by the data-processing inequality,
> $$
> k _ {\rm QAE}(\boldsymbol  {x_i},\boldsymbol {x_j})\geq k(\boldsymbol {x_i},\boldsymbol {x_j}).
> $$
> Consequently, when the kernel is defined via the quantum-state fidelity, the SVM built with the QAE kernel achieves a smaller generalization error than the SVM using the Hamiltonian kernel.

---

> ### Author Response · Authors · 2025-11-28
> **Follow-up Comment**
>
> Dear Reviewer,
>
> I hope this message finds you well. As the discussion period is concluding, we want to ensure that we have addressed all your concerns. If there is anything else you would like us to clarify, please feel free to let us know.
>
> Thank you for your time and effort in reviewing our paper.

---

### Official Review · Reviewer_LbWP · 2025-11-01

**Soundness:** 3
**Presentation:** 3
**Contribution:** 2
**Rating:** 4
**Confidence:** 3

**Summary:**

This paper introduces a quantum autoencoder (QAE) framework for data-driven quantum representation learning, focusing on peptide sequence modelling and classification. The authors argue that most quantum machine learning (QML) pipelines rely on static encodings (e.g., Hamiltonian or angle encodings) that fail to adapt to data structure. The proposed QAE learns compact quantum representations from data in an unsupervised manner, similar to pre-training in classical deep learning.

The paper provides (i) theoretical results on the depth and sample complexity of the variational circuit; (ii) large-scale training of QAEs on 3 million peptide sequences using Hamiltonian-evolved quantum states; and (iii) downstream evaluations on seven peptide classification datasets, comparing learned QAE representations against fixed Hamiltonian baselines.

The results show accuracy improvements ranging from 0.4% to 8.1% over Hamiltonian encodings across tasks, with the learned representations showing transferability without fine-tuning. The study positions quantum autoencoders as an analogue of classical representation learning (e.g., PLMs) within quantum settings .

**Strengths:**

* The paper takes an important step beyond static quantum encodings by treating representation learning as a core problem in quantum pipelines. The analogy to unsupervised pre-training (as in PLMs) is both conceptually interesting and timely for ICLR’s machine learning audience.
* The pre-training on 3 million peptide sequences represents one of the largest QML simulations to date, demonstrating the feasibility of scaling quantum-inspired learning beyond toy datasets.
* Using seven biologically diverse peptide datasets (antihypertensive, antioxidant, BBB-penetrating, cytotoxic, hemolytic, and neurotoxic peptides) provides convincing evidence that the learned quantum representations generalise across domains.

**Weaknesses:**

* The reported improvements (0.4–8.1%) over Hamiltonian baselines are relatively modest compared to PLM gains (1–16%), suggesting limited immediate practical impact. Discussion of why the performance gap is smaller would strengthen the narrative.
* Downstream testing is limited to SVMs with quantum kernels, rather than using QAE features directly for diverse classifiers. While justified by density-matrix representation issues, this restricts insight into how QAEs could integrate into hybrid pipelines.
* The study benchmarks only against Hamiltonian evolution encoding. Other fixed encodings (e.g., amplitude or angle encoding) are not tested, leaving open whether improvements generalise.
* It would be valuable to show which aspects of the learned latent space (e.g., entanglement structure or qubit utilisation) correlate with downstream performance.
* The paper’s comparisons are limited to Hamiltonian-encoded quantum baselines and brief contextual references to PLMs. There are no quantitative benchmarks against classical representation-learning methods such as autoencoders, variational autoencoders (VAEs), or simple self-supervised embeddings trained on the same peptide data. Without such comparisons, it remains unclear whether the observed 0.4–8.1% gains could be replicated by small classical models with similar parameter counts? I'm not wanting to see quantum advantage (which I believe should not be the sole goal of quantum computing), but demonstrating clear empirical advantage to more classical baselines would be almost be essential instead of just comparing to quantum baselines.
* Although the datasets are biologically motivated, the paper does not analyse whether the learned representations correspond to meaningful biochemical properties (e.g., hydrophobicity, charge patterns), which could make the work more relevant to bioinformatics audiences.

**Questions:**

* Do the learned latent representations exhibit measurable structure or interpretability (e.g., clustering by peptide function)?
* How sensitive are the results to the number of qubits

---

> ### Author Response · Authors · 2025-11-21
> **Response to Reviewer LbWP**
>
> Dear Reviewer LbWP:
>
> Thank you for the constructive comments and suggestions. We address each point below with clarifications and supporting results.
> ___
> **Weakness 1**
> > The reported improvements (0.4–8.1%) over Hamiltonian baselines are relatively modest compared to PLM gains (1–16%), suggesting limited immediate practical impact. Discussion of why the performance gap is smaller would strengthen the narrative.
>
> We thank the reviewer for raising this point. We would like to clarify that
> the PLM vs. one-hot comparison is not intended to strengthen any quantum claims, nor to
> imply direct comparability between PLMs and QAEs. Our intention is simply to provide
> context: unsupervised quantum representation learning is still at an early stage, and showing
> the performance of state-of-the-art classical models offers an indirect indication of how much
> progress remains for quantum approaches.
> Within this framing, our objective is not to maximize the numerical performance gap, but
> to demonstrate that the QAE genuinely learns and leads to improvements over a strong
> fixed quantum encoding (Hamiltonian). In this sense, the 0.4–8.1% gains should be interpreted as evidence of successful quantum representation learning, rather than as competitive performance against classical PLMs. The comparison is therefore illustrative rather than competitive, and we have clarified this motivation more clearly in the revised manuscript.
>
> **Weakness 2**
> > Downstream testing is limited to SVMs with quantum kernels, rather than using QAE features directly for diverse classifiers. While justified by density-matrix representation issues, this restricts insight into how QAEs could integrate into hybrid pipelines.
>
>  We agree that exploring hybrid pipelines is an important direction. A possible direction for hybrid pipelines is to consider omitting the partial trace during downstream representation generation. Instead of discarding qubits, one could use the encoder’s full pure output state and extract features directly, either via measurement or classical shadow technique. However, it remains unclear whether bypassing the partial-trace step or applying classical shadows to the resulting pure states would affect performance. Investigating both directions would be an interesting avenue for future work.
>
> **Weakness 3**
> > The study benchmarks only against Hamiltonian evolution encoding. Other fixed encodings (e.g., amplitude or angle encoding) are not tested, leaving open whether improvements generalise.
>
>  To partially address the reviewer’s concern, we have added two more quantum encoding baselines, angle encoding and amplitude encoding, evaluated under identical pipelines (Table 1). These results show that hamiltonian is generally the stronger baseline.
>
> **Table 1.** Comparison across Hamiltonian, Angle Encoding, Amplitude Encoding, and Quantum Autoencoder
> (QAE) using matching qubit counts. Values report mean ± std over 5-fold cross-validation with 20 seeds.
>
> | Dataset                         | Hamiltonian        | Angle              | Amplitude          | QAE         |
> |---------------------------------|--------------------|--------------------|--------------------|---------------------|
> | Antihypertensive                | 0.8768 ± 0.0036    | 0.7902 ± 0.0000    | **0.8969 ± 0.0000**    | 0.8844 ± 0.0024     |
> | Antioxidant                     | 0.8208 ± 0.0004    | 0.7017 ± 0.0759    | 0.5500 ± 0.0000    | **0.8344 ± 0.0079**     |
> | BBB                             | 0.5856 ± 0.0738    | 0.5487 ± 0.0000    | 0.5487 ± 0.0000    | **0.6667 ± 0.0000**     |
> | Cytotoxic                       | 0.7852 ± 0.0000    | 0.5477 ± 0.0000    | 0.5690 ± 0.0156    | **0.7889 ± 0.0000**     |
> | Hemo2                           | 0.6177 ± 0.0653    | 0.5455 ± 0.0000    | 0.5455 ± 0.0000    | **0.6869 ± 0.0000**     |
> | Hemo3                           | 0.7044 ± 0.0000    | 0.5440 ± 0.0000    | 0.6069 ± 0.0000    | **0.7107 ± 0.0000**     |
> | Neurotoxin                      | 0.8871 ± 0.0000    | 0.5258 ± 0.0000    | **0.9410 ± 0.0015**    | 0.9058 ± 0.0019    |
>
> Regarding the suggestions to train QAE models on input states prepared using angle or amplitude encodings, we note that this falls outside the scope of our study. Our goal is to isolate the effect of unsupervised quantum pretraining under a fixed state-preparation model, rather than to explore all possible encoding methods. Training QAEs on alternative encodings would require separate large-scale pretraining runs, and we view this as meaningful but out-of-scope future work rather than essential for the current paper’s main objective that demonstrates QAEs are viable quantum representation learners.

---

> ### Author Response · Authors · 2025-11-21
> **Response to Reviewer LbWP**
>
> **Weakness 4**
> > It would be valuable to show which aspects of the learned latent space (e.g., entanglement structure or qubit utilisation) correlate with downstream performance.
>
> We thank the reviewer for raising this interesting question. Here, we provide a theoretical explanation for why the QAE-transformed latent states lead to improved downstream performance.
>
> Recall that the original quantum kernel function
> $$
> k(\boldsymbol x_i, \boldsymbol x_j) = |\langle \phi(\boldsymbol x_i) \mid \phi(\boldsymbol x_j) \rangle|^2
> = 1 - \mathrm{dist} _ {\mathrm{tr}}^2\left( |\phi(\boldsymbol x_i)\rangle \langle \phi(\boldsymbol x_i)|,|\phi(\boldsymbol x_j) \rangle \langle\phi(\boldsymbol x_j)|  \right),
> $$
>
> where $\mathrm{dist}_{\mathrm{tr}}(\cdot,\cdot)$ denotes the trace distance. As a CPTP map, the QAE approach enables
> the relationship
>
> $$
> \mathrm{dist} _ {\mathrm{tr}}(\phi(\boldsymbol x_i), \phi(\boldsymbol x_j))
> \le
> \mathrm{dist} _ {\mathrm{tr}}\bigl(|\phi(\boldsymbol x_i)\rangle\langle\phi(\boldsymbol x_i)|, |\phi(\boldsymbol x_j)\rangle\langle\phi(\boldsymbol x_j)|\bigr)
> $$
>
> where the quantum feature state is defined as $\phi(\boldsymbol x_i) = \mathrm{QAE}(|\phi(\boldsymbol x_i)\rangle\langle\phi(\boldsymbol x_i)|)$.
> Equivalently, the related quantum kernel function satisfies
> $$
> k_{\mathrm{QAE}}(\boldsymbol x_i, \boldsymbol x_j) \ge k(\boldsymbol x_i, \boldsymbol x_j)
> $$
> for every pair $(i,j)$.
> In the context of the SVM model, the optimal primal value equals the optimal dual value,
> that is
> $$
> p_K = \min_{w,b} \frac{1}{2}\|w\|^2 = \max_{\alpha} W_K(\alpha),
> $$
> where the dual SVM objective for the kernel K is defined by
> $$
> W_K(\alpha) = \sum_i \alpha_i - \frac{1}{2}
> \sum_{i,j} \alpha_i \alpha_j y_i y_j K_{ij}.
> $$
> Fix any parameter $\alpha$ , we consider the difference
> $$
> W_{K_{\mathrm{QAE}}}(\alpha) - W_K(\alpha) =-\frac{1}{2}\sum_{i,j} \alpha_i \alpha_j y_i y_j \Delta_{ij},
> $$
> where $\Delta _ {ij} = [K _ {\mathrm{QAE}}] _ {ij} - [K] _ {ij} \ge 0$. This further results in
> $$
> W_{K_{\mathrm{QAE}}}(\alpha) - W_K(\alpha) \le 0,
> $$
> which directly yields $ p_{K_{\mathrm{QAE}}} \le p_K. $. A standard margin-based generalization upper bound
> states that with probability $\ge 1 - \delta$ over an i.i.d. sample of size $N$, every linear classifier in
> the RKHS with margin $1/pK$ and feature space radius $R$ satisfies
> $$
> \epsilon_{\mathrm{gen}} =
> \mathcal{O}(R^2 p_K / N) + \mathcal{O} \left( \sqrt{\log(1/\delta)/N} \right).
> $$
> This indicates that $\epsilon_{\mathrm{gen}}(K_{\mathrm{QAE}}) \le \epsilon_{\mathrm{gen}}(K)$,  implying QAE method may increase the generalization ability for quantum kernel method. We have supplemented this theoretical analysis in the appendix.

---

> ### Author Response · Authors · 2025-11-21
> **Response to Reviewer LbWP**
>
> **Weakness 5**
> > The paper’s comparisons are limited to Hamiltonian-encoded quantum baselines and brief contextual references to PLMs. There are no quantitative benchmarks against classical representation-learning methods such as autoencoders, variational autoencoders (VAEs), or simple self-supervised embeddings trained on the same peptide data. Without such comparisons, it remains unclear whether the observed 0.4–8.1% gains could be replicated by small classical models with similar parameter counts? I'm not wanting to see quantum advantage (which I believe should not be the sole goal of quantum computing), but demonstrating clear empirical advantage to more classical baselines would be almost be essential instead of just comparing to quantum baselines.
>
> We thank the reviewer for raising this interesting point. To address this, we have added a comparison against classical autoencoders (AE) and variational
> autoencoders (VAE) pretrained on the same 3M peptide sequences and evaluated with
> the same downstream datasets. For completeness, we report two variants: (1) Best
> of six classical classifier give AE/VAE their strongest possible performance (Table 2), and (2) SVM
> only (rbf or linear) to isolate the effect of the learned representation rather than classifier
> choice  (Table 3). These results show that QAE remains competitive with, and often outperforms, classical
> AE/VAE models of comparable parameter scale $(\sim 10^3)$. However, we note that we
> do not draw any claim or conclusion from this comparison, as a fair and systematic analysis
> of classical vs. quantum representation learners would require an extensive study of their
> structures, capacity, and optimization behavior. Our objective here is simply to demonstrate
> that QAEs are viable and effective within the quantum setting, improving over strong fixed
> quantum encodings.
>
> **Table 2.** Comparison of QAE with classical learned representations (AE, VAE) using the **best-performing** classical classifier (selected from six models).
> Values are mean ± std over 5-fold CV with 20 seeds.
>
> | **Dataset**         | **QAE**             | **AE**        | **VAE**       |
> |---------------------|----------------------|------------------------|------------------------|
> | Antihypertensive | **0.8844±0.0024**        | 0.8370±0.0018          | 0.8789±0.0000          |
> | Antioxidant    | **0.8344±0.0079**        | 0.7940±0.0046          | 0.8063±0.0075          |
> | BBB              | 0.6667±0.0000        | **0.7026±0.0000**          | 0.6462±0.0000          |
> | Cytotoxic        | **0.7889±0.0000**        | 0.7023±0.0000          | 0.7148±0.0000          |
> | Hemo2            | 0.6869±0.0000        | **0.6970±0.0000**         | 0.6414±0.0000          |
> | Hemo3            | **0.7107±0.0000**        | 0.6824±0.0000          | 0.6698±0.0000          |
> | Neurotoxin       | **0.9058±0.0019**        | 0.8269±0.0049          | 0.7968±0.0000          |
>
> **Table 3.** Comparison of QAE with classical learned representations (AE, VAE) using classical **SVMs**.
> QAE always uses a trace-distance quantum kernel inside the SVM; therefore its SVM accuracy matches Table 2.
> Values are mean ± std over 5-fold CV with 20 seeds.
>
> | **Dataset**         | **QAE**        | **AE**         | **VAE**        |
> |---------------------|------------------------|------------------------|------------------------|
> | Antihypertensive | **0.8844±0.0024**          | 0.8056±0.0005          | 0.8789±0.0000          |
> | Antioxidant      | **0.8344±0.0079**          | 0.7940±0.0046          | 0.6771±0.0008          |
> | BBB              | **0.6667±0.0000**          | 0.6615±0.0000          | 0.6218±0.0168          |
> | Cytotoxic        | **0.7889±0.0000**          | 0.6972±0.0000          | 0.6300±0.0033          |
> | Hemo2            | **0.6869±0.0000**          | 0.6616±0.0000          | 0.5616±0.0083          |
> | Hemo3           | **0.7107±0.0000**          | 0.6547±0.0019          | 0.6352±0.0000          |
> | Neurotoxin       | **0.9058±0.0019**         | 0.8048±0.0116          | 0.7348±0.0135          |
>
> **Weakness 6**
> > Although the datasets are biologically motivated, the paper does not analyse whether the learned representations correspond to meaningful biochemical properties (e.g., hydrophobicity, charge patterns), which could make the work more relevant to bioinformatics audiences.
>
> In this work, we did not perform a detailed biochemical interpretation analysis, primarily
> because the QAE produces mixed state density matrices after partial trace. To the best of
> our knowledge, there is currently no efficient or standard method to extract interpretable
> biochemical features directly from such quantum representations. Developing tools for interpreting
> mixed quantum states in terms of biochemical properties remains an open problem.
> We believe that this is an important direction, and integrating biochemical interpretability into
> quantum representation learning would be a valuable avenue for future work.

---

> ### Author Response · Authors · 2025-11-21
> **Response to Reviewer LbWP**
>
> **Question 1**
> > Do the learned latent representations exhibit measurable structure or interpretability (e.g., clustering by peptide function)?
>
> We believe this question concerns the same issue of interpretability and measurable structure in the learned quantum representations. Please see our response to **Weakness 6**.
>
> **Question 2**
> > How sensitive are the results to the number of qubits
>
> We performed only a small grid search over QAE architectures: (8 or 10 qubits), (10, 20, or 30 layers) and (1, 2, or 3 traced-out qubits), resulting in 18 configurations in total. Within this search space, we found that downstream performance was not highly sensitive to the number of qubits, both 8 qubits and 10 qubits models achieved the best results on different datasets. Across all datasets, the best performing configurations consistently used 30 layers and no more than 2 traced-out qubits, suggesting that circuit depth and compression strength had a more noticeable impact than input qubit count. We believe this is an interesting future research direction to analyze the effect of learner structure in a more systematic and theoretically guided manner.

---

> ### Author Response · Authors · 2025-11-28
> **Follow-up Comment**
>
> Dear Reviewer,
>
> I hope this message finds you well. As the discussion period is concluding, we want to ensure that we have addressed all your concerns. If there is anything else you would like us to clarify, please feel free to let us know.
>
> Thank you for your time and effort in reviewing our paper.

---

### Official Review · Reviewer_Ff1U · 2025-11-03

**Soundness:** 2
**Presentation:** 2
**Contribution:** 2
**Rating:** 4
**Confidence:** 5

**Summary:**

This paper studies QAEs as learned quantum representations for classical sequence data (peptides). The authors (i) give two theoretical statements: a polylogarithmic depth upper bound for the QAE encoder under local Hamiltonian data encoding, and a uniform-in-training sample complexity bound to estimate the SWAP-test–style loss; (ii) pre-train QAEs on 3M unlabeled peptide sequences encoded via Hamiltonian evolution; and (iii) evaluate the resulting compressed states as features for SVMs with a trace-distance kernel on seven downstream peptide classification tasks. They report +0.4%–8.1% test-accuracy gains over a fixed Hamiltonian encoding baseline.

**Strengths:**

The paper explicitly targets representation learning within QML, comparing learned QAE features against fixed Hamiltonian encodings while holding the downstream classifier constant, which isolates the contribution of representation learning on the quantum side.

In my view, unsupervised training on 3M peptide sequences with small circuits (8–10 qubits, 10–30 layers, and 1–3 traced-out qubits) is a useful empirical data point.

QAEs uniformly outperform the Hamiltonian baseline across seven datasets in radar plots.

The paper candidly notes simulation-only results, kernel-method dependence, and the non-optimality of the Hamiltonian encoding for long sequences.

**Weaknesses:**

The only quantum baseline is the fixed Hamiltonian encoding. While this isolates representation learning, it is a relatively weak comparator. Missing are (a) other learned quantum encoders (e.g., variational feature maps without autoencoding; data-reuploading encoders) and (b) end-to-end quantum classifiers (keeping the same resource envelope) to contextualize the benefits of pretraining. The “PLM vs one-hot” comparison is explicitly not directly comparable and thus does not strengthen the quantum claims (Sec. 4.2, Fig. 4).

The radar plots summarize relative improvements but omit tables of absolute accuracies, standard deviations, and statistical significance across splits. A results table per dataset (mean ± std over 5 folds) would allow readers to judge the practical effect sizes; currently, the gains (0.4%–8.1%) may be within the noise for some datasets.

The path from high-dimensional one-hot peptide vectors to 8–10 qubits via Hamiltonian evolution (Sec. 3.1) compresses massively before autoencoding. It is not obvious how information-preserving or task-aligned this mapping is for long sequences. More ablations are needed: (i) different qubit counts; (ii) alternative encodings (angle/amplitude; learned/Trotterized Hamiltonians or something else); (iii) sensitivity to sequence length thresholds.

All experiments are noiseless simulations; yet the kernel requires estimating the trace distance between reduced density matrices.

Ablations on pretraining value. The claim of “pretrained without fine-tuning” would be stronger if compared against (a) training the same QAE from scratch on each downstream dataset; (b) pretraining on fewer sequences (data-scaling curve); (c) varying the number of traced-out qubits m and depth with fixed compute budget.

**Questions:**

See Weaknesses.

In addition, why fix Hamiltonian evolution as the input encoding before QAE? Can you report results with angle or amplitude encodings (same qubit budget) to show the robustness of the pretraining story?

For a realistic n=10, m∈{1,2,3}, what is the measurement complexity to approximate the trace-distance kernel on current devices? Any preliminary noise simulations (depolarizing/readout) or classical-shadow-based surrogates? Strengthening ablations and reporting are needed.

---

> ### Author Response · Authors · 2025-11-21
> **Response to Reviewer Ff1U**
>
> Dear Reviewer Ff1U:
>
> Thank you for the constructive comments and suggestions. We address each point below with clarifications and supporting results.
> ___
>
> **Weakness 1.1**
> > The only quantum baseline is the fixed Hamiltonian encoding. While this isolates representation learning, it is a relatively weak comparator.
>
> We thank the reviewer for raising this point. Although the reviewer suggests that the Hamiltonian baseline might be a weak comparator,
> our expanded baseline (angle and amplitude encoding; Table 1) shows that the Hamiltonian baseline is, in fact, the stronger overall fixed encoding under the same qubit budget. Hamiltonian encoding outperforms angle encoding on all seven datasets, and outperforms amplitude encoding on five out of seven datasets. While amplitude encoding performs strongly on two datasets (and exceeds QAE on those), Hamiltonian evolution provides the most consistent fixed baseline across tasks, making it an appropriate and competitive reference for evaluating learned representations.
>
> **Table 1.** Comparison across Hamiltonian, Angle Encoding, Amplitude Encoding, and Quantum Autoencoder
> (QAE) (mean ± std over 5-fold cross-validation with 20 seeds).
>
> | Dataset                         | Hamiltonian        | Angle              | Amplitude          | QAE         |
> |---------------------------------|--------------------|--------------------|--------------------|---------------------|
> | Antihypertensive                | 0.8768 ± 0.0036    | 0.7902 ± 0.0000    | **0.8969 ± 0.0000**    | 0.8844 ± 0.0024     |
> | Antioxidant                     | 0.8208 ± 0.0004    | 0.7017 ± 0.0759    | 0.5500 ± 0.0000    | **0.8344 ± 0.0079**     |
> | BBB                             | 0.5856 ± 0.0738    | 0.5487 ± 0.0000    | 0.5487 ± 0.0000    | **0.6667 ± 0.0000**     |
> | Cytotoxic                       | 0.7852 ± 0.0000    | 0.5477 ± 0.0000    | 0.5690 ± 0.0156    | **0.7889 ± 0.0000**     |
> | Hemo2                           | 0.6177 ± 0.0653    | 0.5455 ± 0.0000    | 0.5455 ± 0.0000    | **0.6869 ± 0.0000**     |
> | Hemo3                           | 0.7044 ± 0.0000    | 0.5440 ± 0.0000    | 0.6069 ± 0.0000    | **0.7107 ± 0.0000**     |
> | Neurotoxin                      | 0.8871 ± 0.0000    | 0.5258 ± 0.0000    | **0.9410 ± 0.0015**    | 0.9058 ± 0.0019    |
>
> **Weakness 1.2**
> > Missing are (a) other learned quantum encoders (e.g., variational feature maps without autoencoding; data-reuploading encoders)
>
> Thank you for raising this point. We believe there may be a misunderstanding regarding the role of variational feature-map circuits and data-reuploading architectures.  To the best of our knowledge, variational feature-map circuits and data-reuploading are used as end-to-end variational classifiers rather than as standalone encoders. These models either jointly optimize data encoding gates with trainable parameters under a supervised loss, or repeatedly encode data throughout a variational circuit. In both cases, they do not provide an unsupervised learned representation that can be reused across tasks.  Therefore, we do not view them as directly comparable for our setting. If there are references where these architectures are used purely as encoders, we would appreciate being pointed to them.
>
> **Weakness 1.3**
> > (b) end-to-end quantum classifiers (keeping the same resource envelope) to contextualize the benefits of pretraining.
>
> Thank you for this question. We would like to clarify, both Hamiltonian + kernel and QAE + kernel already constitute standard quantum classifiers, as quantum kernel methods are a widely used and expressive classification approach in prior work. Our aim, however, is not to compare different supervised quantum classifiers, but to evaluate whether unsupervised quantum pretraining can produce more informative representations than fixed encodings. For this purpose, it is necessary to compare the QAE-learned representation against a fixed encoding under the same downstream classifier, which is exactly what we evaluate.
>
> **Weakness 1.4**
> > The “PLM vs one-hot” comparison is explicitly not directly comparable and thus does not strengthen the quantum claims (Sec. 4.2, Fig. 4).
>
> The PLM vs. one-hot comparison is not meant to strengthen the quantum claims or to suggest direct comparability between PLMs and QAEs. Our intention is simply to provide context: unsupervised quantum representation learning is still at an early stage, and showing the performance of state-of-the-art classical models offers an indirect sense of how much progress the quantum approach still needs to make. This comparison is therefore illustrative rather than competitive. As requested by another reviewer, we now include classical autoencoder and variational autoencoder with similar parameter counts in the revised manuscript and in our responses to Reviewer *LbWP*, but our quantum specific claims continue to rely solely on within quantum comparisons (fixed Hamiltonian encoding versus the learned QAE representation).

---

> ### Author Response · Authors · 2025-11-21
> **Response to Reviewer Ff1U**
>
> **Weakness 2**
> > The radar plots summarize relative improvements but omit tables of absolute accuracies, standard deviations, and statistical significance across splits. A results table per dataset (mean ± std over 5 folds) would allow readers to judge the practical effect sizes; currently, the gains (0.4%–8.1%) may be within the noise for some datasets.
>
> Thank you for raising this point. We now include the full per-dataset table corresponding exactly to the values shown in the radar plot, reporting the mean accuracies together with standard deviations over 5-fold cross-validation and 20 random seeds (Table 2). This provides the absolute effect sizes requested by the reviewer and allows a clearer assessment of the practical significance of the observed gains.
>
> **Table 2. Per-dataset comparison between the highest-accuracy QAE configuration and the Hamiltonian baseline (Mean ± std over 5-fold CV with 20 random seeds. QAE Config: L denotes the number of layers, and T denotes the number of traced-out qubits.).**
>
> | Dataset          | # Qubits | QAE Config | Hamiltonian  | QAE | Δ |
> | ---------------- | -------- | -------------- | ------------------------ | ------------------- | ------- |
> | Antihypertensive | 10       | 30L-1T           | 0.8768 ± 0.0036          | **0.8844 ± 0.0024** | +0.0076 |
> | Antioxidant      | 8        | 30L-2T           | 0.8208 ± 0.0004          | **0.8344 ± 0.0079** | +0.0136 |
> | BBB-penetrating  | 8        | 30L-2T           | 0.5856 ± 0.0738          | **0.6667 ± 0.0000** | +0.0810 |
> | Cytotoxic        | 8        | 30L-1T           | 0.7852 ± 0.0000          | **0.7889 ± 0.0000** | +0.0038 |
> | Hemo2            | 8        | 30L-2T           | 0.6177 ± 0.0653          | **0.6869 ± 0.0000** | +0.0692 |
> | Hemo3            | 8        | 30L-1T           | 0.7044 ± 0.0000          | **0.7107 ± 0.0000** | +0.0063 |
> | Neurotoxin       | 10       | 30L-2T           | 0.8871 ± 0.0000          | **0.9058 ± 0.0019** | +0.0187 |
>
> **Weakness 3**
> > The path from high-dimensional one-hot peptide vectors to 8–10 qubits via Hamiltonian evolution (Sec. 3.1) compresses massively before autoencoding. It is not obvious how information-preserving or task-aligned this mapping is for long sequences. More ablations are needed: (i) different qubit counts; (ii) alternative encodings (angle/amplitude; learned/Trotterized Hamiltonians or something else); (iii) sensitivity to sequence length thresholds.
>
> Thank you for this detailed suggestions.
>
> (i) Regarding qubit counts, Table 1 shows that both 8-qubit and 10-qubit QAEs achieve the best performance on different datasets: 10-qubit models perform best on five out of seven tasks, while 8-qubit models perform best on the remaining two. All of our best performing QAEs use relatively expressive circuits (30 layers and at most two qubits traced out), suggesting that models with slightly larger latent spaces and deeper circuits can yield higher-quality representations. However, our focus is not to identify the optimal architecture, but to demonstrate that the QAE consistently learns useful representations across datasets and qubit budgets. A systematic exploration of larger circuits is an interesting direction for future work but lies outside the scope of the present study.
>
> (ii) For the alternative-encoding ablation, our understanding is that the reviewer is suggesting using angle, amplitude, or other encodings as the input states to the QAE itself. This falls outside the scope of the present work: our goal is to isolate the effect of unsupervised representation learning under a fixed state-preparation model, rather than to exhaustively explore alternative input encodings for the QAE.
>
> (iii) Regarding sensitivity to sequence-length thresholds, we first examined the empirical length distribution of each dataset and removed only the extremely long sequences that occur very rarely. After this filtering step, all remaining peptides are padded with zeros to match the length of the longest retained sequence, and the resulting fixed-size vectors are then flattened into one-hot encodings. Thus, the threshold only determines which rare outliers are removed; it does not affect the sequential structure seen by the model. Because the flattened one-hot representations are highly sparse and dominated by local residue identity rather than absolute length, small adjustments to this threshold primarily change the number of padding positions and do not materially affect the encoded quantum states. Our focus is therefore on representation learning under a fixed and consistent preprocessing pipeline, and a systematic study of different threshold choices lies outside the scope of the present work.

---

> ### Author Response · Authors · 2025-11-21
> **Response to Reviewer Ff1U**
>
> **Weakness 4**
> > All experiments are noiseless simulations; yet the kernel requires estimating the trace distance between reduced density matrices.
>
> The reviewer is correct that trace distance estimation can be sensitive to noise on real hardware. To address this point, we have added a depolarizing noise study (Table 3) evaluating both the Hamiltonian and QAE representations under noise levels
> $p \in$ {0.00,0.02,0.05,0.10,0.20} . Overall, the QAE maintains its advantage at low and moderate noise levels, and for 4 out of 7 datasets it continues to outperform the Hamiltonian baseline across the full noise range. For the remaining datasets, the Hamiltonian encoding becomes competitive or slightly better at higher noise levels (e.g.,  $p$ = 0.10 and $p$=0.20), and in two cases its accuracy even increases slightly under higher noise. Overall, both methods degrade only gradually, and the QAE does not collapse under noise. These results provide preliminary evidence that the learned representations are reasonably robust and that the observed improvements are not solely artifacts of noiseless simulation.
>
> **Table 3. Noise robustness under depolarizing noise.**
>
> | Dataset      | Method      | p=0.00 | p=0.02 | p=0.05 | p=0.10 | p=0.20 |
> |--------------|-------------|--------|--------|--------|--------|--------|
> | Antihypertensive | QAE         | **0.8844** | **0.8825** | **0.8825** | **0.8832** | **0.8799** |
> |              | Hamiltonian | 0.8768 | 0.8773 | 0.8789 | 0.8773 | 0.8765 |
> | Antioxidant  | QAE         | **0.8344** | **0.8360** | **0.8379** | **0.8345** | **0.8345** |
> |              | Hamiltonian | 0.8208 | 0.8224 | 0.8241 | 0.8241 | 0.8214 |
> | BBB          | QAE         | **0.6667** | **0.6667** | **0.6615** |0.6564 | 0.6462 |
> |              | Hamiltonian | 0.5856 | 0.6010 | 0.6226 | **0.6800** | **0.6826** |
> | Cytotoxic    | QAE         | **0.7889** | **0.7902** | **0.7902** | **0.7877** | **0.7876** |
> |              | Hamiltonian | 0.7852 | 0.7839 | 0.7839 | 0.7839 | 0.7827 |
> | Hemo2        | QAE         | **0.6869** | **0.6818** | **0.6818** | **0.6566** | 0.6162|
> |              | Hamiltonian | 0.6177 | 0.6177 | 0.6308 | 0.6308 | **0.6465** |
> | Hemo3        | QAE         | **0.7107** | **0.7107** | **0.7107** | **0.7013** | 0.6824 |
> |              | Hamiltonian | 0.7044 | 0.7044 | 0.7075 | **0.7013** | **0.6887** |
> | Neurotoxin   | QAE         | **0.9058** | **0.9058** | **0.9045** | **0.9055** | **0.9018** |
> |              | Hamiltonian | 0.8871 | 0.8887 | 0.8871 | 0.8871 | 0.8913 |
>
> **Weakness 5.1**
> > Ablations on pretraining value. The claim of “pretrained without fine-tuning” would be stronger if compared against (a) training the same QAE from scratch on each downstream dataset
>
> We thank the reviewer for this detailed suggestions.
>
> We have added the ablation requested in (a): we train the
> same QAE architecture from scratch on each downstream dataset (Table 4). The pretrained
> QAE consistently outperforms the downstream-trained variant across all seven tasks.
>
> **Table 4.** Per-dataset comparison between the highest-accuracy QAE configuration,
> the Hamiltonian baseline, and the same QAE architecture trained from scratch on each
> downstream dataset (mean ± std over 5-fold CV with 20 seeds).
>
> | Dataset           | Hamiltonian        | QAE (Pretrained)     | QAE (Downstream-trained) |
> |-------------------|--------------------|-----------------------|---------------------------|
> | Antihypertensive  | 0.8768 ± 0.0036    | **0.8844 ± 0.0024**       | 0.8813 ± 0.0000           |
> | Antioxidant       | 0.8208 ± 0.0004    | **0.8344 ± 0.0079**       | 0.7955 ± 0.0025           |
> | BBB               | 0.5856 ± 0.0738    | **0.6667 ± 0.0000**       | 0.6308 ± 0.0000           |
> | Cytotoxic         | 0.7852 ± 0.0000    | **0.7889 ± 0.0000**       | 0.7764 ± 0.0000           |
> | Hemo2             | 0.6177 ± 0.0653    | **0.6869 ± 0.0000**       | 0.6558 ± 0.0253           |
> | Hemo3             | 0.7044 ± 0.0000    | **0.7107 ± 0.0000**       | 0.7013 ± 0.0000           |
> | Neurotoxin        | 0.8871 ± 0.0000    | **0.9058 ± 0.0019**       | 0.8890 ± 0.0039           |
>
> **Weakness 5.2**
>  > (b) pretraining on fewer sequences (data-scaling curve);
>
> (b) We agree that a data scaling curve for pretraining would be valuable in its own right. However, we believe it addresses a different question from the one we focus on in this work: our aim is to show that unsupervised quantum pretraining can yield transferable representations on real-world datasets, rather than to study how performance changes as the size of the pretraining corpus is varied. A detailed scaling analysis would broaden the scope considerably, and we view it as a meaningful direction for future work.

---

> ### Author Response · Authors · 2025-11-21
> **Response to Reviewer Ff1U**
>
> **Weakness 5.3**
> > (c) varying the number of traced-out qubits m and depth with fixed compute budget.
>
> (c) This architectural ablation was already performed as part of the model search in the original submission. We evaluated multiple combinations of circuit depth (10, 20, and 30 layers) and numbers of traced out qubits $m \in$ {1,2,3} , and selected the best configuration for each downstream dataset. The full search results are provided in the anonymous GitHub repository linked in the submission, but were not included in the paper itself to avoid shifting the focus toward architecture optimization rather than representation learning. With this relatively small search grid, we did not observe a clear trend across depth or $m$. We note, however, that the highest-performing QAEs for most datasets happened to use deeper circuits (30 layers) with at most two traced-out qubits (as shown in Table 2 above), suggesting that moderately expressive models can be beneficial in some cases. A more extensive architectural study would be required to draw firm conclusions, and falls outside the scope of the present work. We view this as a natural direction for future investigation, especially if theoretical guidance on QAE architecture becomes available.
>
> **Question 1**
> > In addition, why fix Hamiltonian evolution as the input encoding before QAE? Can you report results with angle or amplitude encodings (same qubit budget) to show the robustness of the pretraining story?
>
> Thank you for this question. As discussed in our response to Weakness 1.1, we selected Hamiltonian evolution as the input encoding because it is the stronger and consistent fixed baseline, outperforming angle encoding on all datasets and amplitude encoding on five out of seven datasets (Table 1).
>
> Regarding the suggestion to train QAE models on input states prepared using angle or amplitude encodings, this falls outside the scope of our study. Our goal is to isolate the effect of unsupervised quantum pretraining under a fixed state-preparation model, rather than to explore all possible encoding methods. Training QAEs on alternative encodings would require separate large-scale pretraining runs, and we view this as meaningful but out-of-scope future work rather than essential for the current paper’s main objective that demonstrates QAEs are viable quantum representation learners.
>
> **Question 2**
> > For a realistic $n$ = 10, $m \in$ {1,2,3}, what is the measurement complexity to approximate the trace-distance kernel on current devices?
>
> In our manuscript, we state that the trace distance can be efficiently estimated with the sample complexity of $\mathcal{O}({\rm rank}^2(\phi ( \boldsymbol {x_j} ))/\epsilon^5)$, where $\epsilon$ represents the additive error, and ${\rm rank}(\phi(\boldsymbol x_j))=\mathcal{O}(2^m)$. In the numerical simulation, we set $\epsilon=10^{-4}$ and ${\rm rank}(\phi(\boldsymbol x_j))=2^m$.
>
> **Question 3**
> > Any preliminary noise simulations (depolarizing/readout) or classical-shadow-based surrogates? Strengthening ablations and reporting are needed.
>
> We have included a depolarizing noise simulation, please see our response to Weakness 4 for details and results.

---

> ### Author Response · Authors · 2025-11-28
> **Follow-up Comment**
>
> Dear Reviewer,
>
> I hope this message finds you well. As the discussion period is concluding, we want to ensure that we have addressed all your concerns. If there is anything else you would like us to clarify, please feel free to let us know.
>
> Thank you for your time and effort in reviewing our paper.

---

### Official Review · Reviewer_oEKu · 2025-11-07

**Soundness:** 3
**Presentation:** 3
**Contribution:** 2
**Rating:** 4
**Confidence:** 3

**Summary:**

This paper studies quantum autoencoders (QAEs) as a method for unsupervised quantum representation learning for classical peptide sequences. The authors pre-train QAEs on 3 million peptide samples encoded via Hamiltonian evolution, producing compressed quantum states used as representations. The learned representations are evaluated on seven downstream peptide classification datasets using quantum kernels with SVM classifiers. Compared against fixed Hamiltonian-evolution encodings, the QAE-derived representations improve test accuracy by 0.4–8.1%. The authors also provide theoretical results claiming (1) polylogarithmic circuit depth is sufficient for implementing QAEs under local Hamiltonians, and (2) efficient sample complexity for estimating the QAE loss throughout training. The work emphasizes that the goal is not quantum advantage but establishing QAEs as effective, scalable quantum representation learners on large classical datasets.

**Strengths:**

- The pre-training on 3 million peptide sequences is a significant step up from the toy problems (e.g., MNIST, Iris) often seen in QML literature. This demonstrates an attempt to scale these methods to a problem of practical and real-world size, which is conceptually aligned with classical deep representation learning.

- The comparative framework (in Figure 1) is well-designed. By having both the QAE pipeline and the baseline pipeline start with the same Hamiltonian encoding, the experiment effectively isolates the contribution of the QAE's learned transformation.

- The paper provides two theoretical results: polylogarithmic depth for implementing QAEs under specific Hamiltonian assumptions, and a sample-complexity bound via local scrambling and classical shadows. Although high-level, these results offer theoretical grounding for scalability.

**Weaknesses:**

- The primary empirical evidence for the QAE's utility is an accuracy gain of 0.4% to 8.1%. On several datasets, the gains are marginal (e.g., 0.4%, 0.6%, 0.8%, 1.4%). These small improvements make it unclear about the practical significance of the approach, especially when considering the overhead of pre-training the QAE. Also the experiments are conducted entirely in a noiseless simulation. Since variational quantum algorithms are sensitive to hardware noise, a gain of 0.4% or 0.6% would almost be erased by any realistic noise model. Without at least a preliminary analysis of noise robustness, the paper's claim of "viability for practical quantum applications" feels a bit unsubstantiated.

- The use of constant-weight Pauli Hamiltonians, locality conditions, and scrambling assumptions for two-qubit gates are strong constraints. It is unclear how these results translate to realistic QAEs used in the experiments. The connection between the theoretical analysis and the specific circuits used is not fully convincing, partiuclarly in real world applications.

- Only Hamiltonian evolution is used as baseline. Other natural baselines—angle encoding, amplitude encoding, shallow variational encoders, or random features—are omitted.

**Questions:**

- How sensitive are the reported gains to the choice of kernel? Would fidelity kernels or projected kernels produce similar improvements?

- Why is the baseline restricted to Hamiltonian evolution? Could the authors include comparisons against at least angle encoding or random variational encodings?

- Given the small performance margins, how do you expect these gains to hold up under realistic noise? Even a simple depolarizing noise simulation would add significant weight to your claims of practical viability.

---

> ### Author Response · Authors · 2025-11-21
> **Response to Reviewer oEKu**
>
> **Dear Reviewer oEKu:**
>
> Thank you for the constructive comments and suggestions. We address each point below with clarifications and supporting results.
> ___
> **Weakness 1.1**
> > The gains are marginal and the practical significance is unclear
>
> We agree that some of the accuracy gains are marginal, and to make this clear, we included
> Table 1, which reports the exact mean ± std values for the best quantum autoencoders (QAE) identified through
> cross-validation. These results show that, while the gains are small in some datasets, they are
> consistent and the variance is low, reflecting stable improvements over the baseline. Table 1
> corresponds exactly to the radar plots in the main text. It is also important to note that the
> Hamiltonian evolution encoding is already a strong and competitive baseline, as seen in Table
> 3 below, which makes it a challenging starting point for improvement. Our focus in this work is on
> whether unsupervised quantum representation learning can provide improvements over fixed
> encodings, rather than on how large the improvement can be, especially given that we explore
> only a small subset of possible QAE architectures. Although some accuracy gains are modest,
> we believe the results reveal a viable research direction for learned quantum representations
> applied to classical data.
>
> **Table 1. Per-dataset comparison between the highest-accuracy QAE configuration and the Hamiltonian baseline (Mean ± std over 5-fold CV with 20 random seeds. QAE Config: L denotes the number of layers, and T denotes the number of traced-out qubits.).**
>
>
> | Dataset          | # Qubits | QAE Config | Hamiltonian  | QAE | Δ |
> | ---------------- | -------- | -------------- | ------------------------ | ------------------- | ------- |
> | Antihypertensive | 10       | 30L-1T           | 0.8768 ± 0.0036          | **0.8844 ± 0.0024** | +0.0076 |
> | Antioxidant      | 8        | 30L-2T            | 0.8208 ± 0.0004          | **0.8344 ± 0.0079** | +0.0136 |
> | BBB-penetrating  | 8        | 30L-2T           | 0.5856 ± 0.0738          | **0.6667 ± 0.0000** | +0.0810 |
> | Cytotoxic        | 8        | 30L-1T           | 0.7852 ± 0.0000          | **0.7889 ± 0.0000** | +0.0038 |
> | Hemo2            | 8        | 30L-2T           | 0.6177 ± 0.0653          | **0.6869 ± 0.0000** | +0.0692 |
> | Hemo3            | 8        | 30L-1T           | 0.7044 ± 0.0000          | **0.7107 ± 0.0000** | +0.0063 |
> | Neurotoxin       | 10       | 30L-2T           | 0.8871 ± 0.0000          | **0.9058 ± 0.0019** | +0.0187 |
>
>
> **Weakness 1.2**
> > Since variational quantum algorithms are sensitive to hardware noise, a gain of 0.4% or 0.6% would almost be erased by any realistic noise model. Without at least a preliminary analysis of noise robustness, the paper's claim of "viability for practical quantum applications" feels a bit unsubstantiated.
>
> Thank you for raising this point. To address this point, we have added a depolarizing noise simulation (Table 2) covering noise levels p ∈ {0.00, 0.02, 0.05, 0.10, 0.20}. The results show that QAE representations remain stable under moderate noise and, in most datasets, continue to outperform or match the Hamiltonian baseline across all noise levels. Interestingly, in a few cases, the Hamiltonian baseline improves slightly with noise, likely due to noise acting as a regularizer. Overall, the noise study indicates that the observed gains are not eliminated by realistic levels of noise and that both representations maintain comparable robustness.
>
> **Table 2. Noise robustness under depolarizing noise.**
>
> | Dataset      | Method      | p=0.00 | p=0.02 | p=0.05 | p=0.10 | p=0.20 |
> |--------------|-------------|--------|--------|--------|--------|--------|
> | Antihypertensive | QAE         | **0.8844** | **0.8825** | **0.8825** | **0.8832** | **0.8799** |
> |              | Hamiltonian | 0.8768 | 0.8773 | 0.8789 | 0.8773 | 0.8765 |
> | Antioxidant  | QAE         | **0.8344** | **0.8360** | **0.8379** | **0.8345** | **0.8345** |
> |              | Hamiltonian | 0.8208 | 0.8224 | 0.8241 | 0.8241 | 0.8214 |
> | BBB          | QAE         | **0.6667** | **0.6667** | **0.6615** |0.6564 | 0.6462 |
> |              | Hamiltonian | 0.5856 | 0.6010 | 0.6226 | **0.6800** | **0.6826** |
> | Cytotoxic    | QAE         | **0.7889** | **0.7902** | **0.7902** | **0.7877** | **0.7876** |
> |              | Hamiltonian | 0.7852 | 0.7839 | 0.7839 | 0.7839 | 0.7827 |
> | Hemo2        | QAE         | **0.6869** | **0.6818** | **0.6818** | **0.6566** | 0.6162|
> |              | Hamiltonian | 0.6177 | 0.6177 | 0.6308 | 0.6308 | **0.6465** |
> | Hemo3        | QAE         | **0.7107** | **0.7107** | **0.7107** | **0.7013** | 0.6824 |
> |              | Hamiltonian | 0.7044 | 0.7044 | 0.7075 | **0.7013** | **0.6887** |
> | Neurotoxin   | QAE         | **0.9058** | **0.9058** | **0.9045** | **0.9055** | **0.9018** |
> |              | Hamiltonian | 0.8871 | 0.8887 | 0.8871 | 0.8871 | 0.8913 |

---

> ### Author Response · Authors · 2025-11-21
> **Response to Reviewer oEKu**
>
> **Weakness 2**
> > The use of constant-weight Pauli Hamiltonians, locality conditions, and scrambling assumptions for two-qubit gates are strong constraints. It is unclear how these results translate to realistic QAEs used in the experiments. The connection between the theoretical analysis and the specific circuits used is not fully convincing, partiuclarly in real world applications.
>
> We thank the reviewer for this interesting question. Here, the locally scrambling property represents a distribution that is invariant under rotation by random single-qubit Clifford gates, which can be thought of as ensembles of quantum circuits that look random locally on each qubit. We note that the locally scrambling property is satisfied by a wide range of deep and shallow unstructured parameterised quantum circuits of different topologies currently used by variational quantum algorithms [1,2,3]. In the unsupervised learning phase, the QAE method utilizes a variational quantum circuit, and this thus can be considered as a quantum approach where each local gate satisfies fast scrambling property.
>
> **Weakness 3**
> > Only Hamiltonian evolution is used as baseline. Other natural baselines—angle encoding, amplitude encoding, shallow variational encoders, or random features—are omitted.
>
> We thank the reviewer for this suggestion. We have included
> a table with additional quantum baselines (Table 3) to provide a broader comparison. These
> results show that, in general, the Hamiltonian evolution encoding is a stronger and more
> competitive baseline than the alternative fixed encodings considered, making it an appropriate
> reference point for evaluating whether learned representations provide any benefit.
>
> **Table 3.** Comparison across Hamiltonian, Angle Encoding, Amplitude Encoding, and Quantum Autoencoder
> (QAE) using matching qubit counts. Values report mean ± std over 5-fold cross-validation with 20 seeds.
>
> | Dataset                         | Hamiltonian        | Angle              | Amplitude          | QAE         |
> |---------------------------------|--------------------|--------------------|--------------------|---------------------|
> | Antihypertensive                | 0.8768 ± 0.0036    | 0.7902 ± 0.0000    | **0.8969 ± 0.0000**    | 0.8844 ± 0.0024     |
> | Antioxidant                     | 0.8208 ± 0.0004    | 0.7017 ± 0.0759    | 0.5500 ± 0.0000    | **0.8344 ± 0.0079**     |
> | BBB                             | 0.5856 ± 0.0738    | 0.5487 ± 0.0000    | 0.5487 ± 0.0000    | **0.6667 ± 0.0000**     |
> | Cytotoxic                       | 0.7852 ± 0.0000    | 0.5477 ± 0.0000    | 0.5690 ± 0.0156    | **0.7889 ± 0.0000**     |
> | Hemo2                           | 0.6177 ± 0.0653    | 0.5455 ± 0.0000    | 0.5455 ± 0.0000    | **0.6869 ± 0.0000**     |
> | Hemo3                           | 0.7044 ± 0.0000    | 0.5440 ± 0.0000    | 0.6069 ± 0.0000    | **0.7107 ± 0.0000**     |
> | Neurotoxin                      | 0.8871 ± 0.0000    | 0.5258 ± 0.0000    | **0.9410 ± 0.0015**    | 0.9058 ± 0.0019    |
>
> **Question 1**
> > How sensitive are the reported gains to the choice of kernel? Would fidelity kernels or projected kernels produce similar improvements?
>
> Thank you for this thoughtful question. Our manuscript studies how the QAE method enhances fidelity kernels.
> The projected kernel  $Tr [Tr_A [\psi(x_i)] Tr_A [\psi(x_j)]]$
>  ,  where  $A$  represents a qubit index $A\in[n]$, is a special case of the QAE-fidelity kernel. Unlike the QAE-fidelity kernel, the projected kernel does not extract features via an unsupervised learning procedure (as QAE does). As a result, the QAE-fidelity kernel function should outperform the standard projected kernel function.
>
>
> **Question 2**
> >Why is the baseline restricted to Hamiltonian evolution? Could the authors
> include comparisons against at least angle encoding or random variational
> encodings?
>
> Please see our response to Weakness 3.
>
> **Question 3**
> >Given the small performance margins, how do you expect these gains to hold up under realistic noise? Even a simple depolarizing noise simulation would add significant weight to your claims of practical viability.
>
> Please see our response to Weakness 1.2.
>
> **Reference:**
>
> [1] Cerezo, M., Sone, A., Volkoff, T., Cincio, L., & Coles, P. J. (2021). Cost function dependent barren plateaus in shallow parametrized quantum circuits. Nature communications, 12(1), 1791.
>
> [2] Kandala, A., Mezzacapo, A., Temme, K., Takita, M., Brink, M., Chow, J. M., & Gambetta, J. M. (2017). Hardware-efficient variational quantum eigensolver for small molecules and quantum magnets. nature, 549(7671), 242-246.
>
> [3] Angrisani, A., Schmidhuber, A., Rudolph, M. S., Cerezo, M., Holmes, Z., & Huang, H. Y. (2024). Classically estimating observables of noiseless quantum circuits. arXiv preprint arXiv:2409.01706.

---

> ### Author Response · Authors · 2025-11-28
> **Follow-up Comment**
>
> Dear Reviewer,
>
> I hope this message finds you well. As the discussion period is concluding, we want to ensure that we have addressed all your concerns. If there is anything else you would like us to clarify, please feel free to let us know.
>
> Thank you for your time and effort in reviewing our paper.

---

> ### Comment · Reviewer_oEKu · 2025-11-28
> **Reply to the authors**
>
> I thank the authors for the supplementary simulations and the revision addressing some of my concerns. However, my primary reservation remains: the proposed QAE demonstrates only marginal improvement over standard quantum encoding methods. Given that the main bottleneck in current QML research is not data encoding performance, such an improvement offers limited impact on the practical viability of QML. Therefore, I will maintain my original score.

---

> > ### Author Response · Authors · 2025-11-30
> > **Respones to Reviewer oEKu**
> >
> > We appreciate the reviewer’s concern regarding the practical impact of improving quantum encodings.  We would like to briefly emphasize why improving quantum encodings is meaningful.
> > Recent theoretical work shows that the encoding determines whether a QML model can exhibit any quantum advantage at all : well structured encodings can yield provable separations [4,5], while poorly designed or random encodings make the model classically simulable or collapse the kernel to exponentially small values, eliminating any benefit [6,7,8]. Designing effective encodings is therefore a central challenge in QML.
> >
> > In this context, our work provides both empirical (3M sequences) and theoretical evidence that a QAE can learn a better encoding than a fixed hand-crafted map. Even though the improvements are moderate under a highly compressed 8–10 qubit setting, the result is significant because it demonstrates that quantum encodings can be improved via data-driven learning, rather than being manually specified.
> >
> > Finally, this paper aims to open up a viable research direction, analogous to early work on classical autoencoders and word embeddings, where initial gains were modest but enabled scalable representation learning frameworks. Given the field’s early stage, we believe even modest improvements are informative.
> >
> > We have updated the revised manuscript to reflect this motivation more explicitly.
> >
> >
> > **Reference:**
> >
> > [4] Liu, Y., Arunachalam, S., & Temme, K. (2021). A rigorous and robust quantum speed-up in supervised machine learning. Nature Physics, 17(9), 1013-1017.
> >
> > [5] Huang, H. Y., Broughton, M., Mohseni, M., Babbush, R., Boixo, S., Neven, H., & McClean, J. R. (2021). Power of data in quantum machine learning. Nature communications, 12(1), 2631.
> >
> > [6] Angrisani, A., Schmidhuber, A., Rudolph, M. S., Cerezo, M., Holmes, Z., & Huang, H. Y. (2024). Classically estimating observables of noiseless quantum circuits. arXiv preprint arXiv:2409.01706.
> >
> > [7] Bermejo, P., Braccia, P., Rudolph, M. S., Holmes, Z., Cincio, L., & Cerezo, M. (2024). Quantum convolutional neural networks are (effectively) classically simulable. arXiv preprint arXiv:2408.12739.
> >
> > [8] Thanasilp, S., Wang, S., Cerezo, M., & Holmes, Z. (2024). Exponential concentration in quantum kernel methods. Nature communications, 15(1), 5200.

---

### Author Response · Authors · 2025-11-30
**Author Summary for the Area Chair**

We thank the Area Chair for their time and effort in evaluating our submission, and we appreciate the reviewers for their thoughtful feedback. In the revised manuscript and rebuttal, we have addressed all technical, empirical, and clarity related concerns raised by the four reviewers.
___
 Below is a brief summary of the major revisions:

**1. Clarified the importance of encoding in QML.**
We added a concise discussion in the Introduction explaining why data encoding is a key factor determining whether QML models can achieve any practical or theoretical advantage, supported by recent literature. This highlights the conceptual motivation behind learning improved quantum encodings through QAEs.

**2. Expanded empirical evaluation.**

- Added mean ± std for all datasets across 5-fold CV with 20 seeds.

- Included depolarizing noise simulations showing that QAE improvements are stable under realistic noise levels.

- Added additional quantum baselines (angle encoding and amplitude encoding), demonstrating that the Hamiltonian baseline is nontrivial and competitive.

- Added classical AE and VAE baselines to contextualize performance relative to classical representation learning.

- Added QAE pretraining vs. downstream-only training ablations, showing clear benefit from large-scale unsupervised pretraining.

**3. Improved theoretical positioning.**
We clarified the connection between the theoretical results and the empirical setting, explaining how the QAE (as a CPTP map) can reduce trace distance and tighten generalization bounds for kernel-based classifiers.

**4. Improved presentation and fairness.**
We clarified that both Hamiltonian and QAE representations use the same trace-distance kernel, ensuring fairness in comparison. Several sections have been reorganized for improved clarity.


**5. Updated manuscript text and captions.**
We revised several parts of the manuscript for clarity, including updating the radar plot caption and added new text to reflect the newly added tables and analyses introduced in the revision.

Overall, the revision adds clarity to the methodology, expands the empirical evaluation, and positions the work more clearly as a first step toward data-driven quantum encoding design. We again thank the Area Chair for overseeing the review process and for their time in evaluating our work.

---

### Meta-Review · Area_Chair_pVid · 2026-01-02

**Summary:**

The authors present a method for using quantum ML to come up with better features for classification problems involving challenging computational chemistry problems related to peptides. The method presents a small improvement over baselines and the features are generally fed into kernel SVMs for evaluation. Many of the reviewers had concerns about the baselines, which only included non-deep-learning methods and did not include vanilla autoencoders or variational autoencoders, but the authors addressed this in the rebuttal. Reviewers also pointed out that features derived from protein language models seemed to do better than those from the quantum autoencoder. Overall the paper seems to show marginal gains but not enough to justify its use relative to protein language model features, and does not provide any evidence that these gains will improve as the method is scaled to true quantum hardware. I concur with the reviewers and recommend against acceptance.

**Reviewer Concerns:**

The authors adequately addressed the reviewer concerns about lack of comparison against classical autoencoders. The broader concerns about the small magnitude of gains and the better performance of PLMs were not addressed.

**Reviewer Scores:**

The first reviewer responded and did not raise their score. Some of the other reviewers might have raised their scores, but the overall ratings were not strong enough for it to make a difference in the decision.

---

### Decision · Program_Chairs · 2026-01-26

Reject